# All-you-can-eat buffet: A spider-specialized bat species (*Myotis emarginatus*) turns into a pest fly eater around cattle

**Chloé Vescera**[1][¤]*, **Cécile Van Vyve**[2][‡], **Quentin Smits**[3][‡], **Johan R. Michaux**[1]

**1** Conservation Genetics Laboratory, University of Liège, Liège, Belgium, **2** Département Etudes—Plecotus, Natagora, Namur, Belgium, **3** Département de l'Etude du Milieu Naturel et Agricole (DEMNA), Service Public de Wallonie (SPW), Gembloux, Belgium

☉ These authors contributed equally to this work.
¤ Current address: Conservation Genetics Laboratory, Institut de Botanique, Quartier Vallée, Chemin de la Vallée, Liège, Belgium
‡ These authors also contributed equally to this work.
* chloe.vescera@doct.uliege.be

**Data Availability Statement:** Raw DNA sequences of prey are available on Sequence Read Archive (SRA) public repository under the accession ID PRJNA1019357. Fasta sequences attributed to

## Abstract

Determining the dietary spectrum of European insectivorous bats over time is the cornerstone of their conservation, as it will aid our understanding of foraging behavior plasticity in response to plummeting insect populations. Despite the global decline in insects, a restricted number of arthropod pest species thrive. Yet past research has overlooked the potential of European bats to suppress pests harmful to woodlands or livestock, in spite of their economic relevance. Here we investigated the diet composition, its breeding season variations and pest consumption of an insectivorous bat species (*Myotis emarginatus*), at the northern edge of its range (Wallonia, Belgium). We also explored the prey ecology to gain insight into the hunting strategies and foraging habitats of this bat species. We used DNA metabarcoding to amplify two COI markers within 195 bat droppings collected in June, July and August, thereby identifying 512 prey taxa predominated by Diptera, Araneae and Lepidoptera. Overall, in 97% of the samples we detected at least one of the 58 potential pest taxa, 41 of which targeting trees. The June samples were marked by a diet rich in orb-weaver spiders, in accordance with the archetypal diet of *M. emarginatus* bats. However, during the highly energy demanding July-August parturition and lactation period, roughly 55% of the dropping samples contained two cattle fly pests (*Stomoxys calcitrans* and *Musca domestica)*. Moreover, among the 88 Diptera species preyed upon by *M. emarginatus* in July and August, these flies accounted for around 50% of the taxa occurrences. This plasticity—the switch from a spider-rich to a fly-rich diet—seems providential considering the dramatic ongoing drop in insect populations but this involves ensuring bat-friendly cattle farming. Our results revealed that bats widely consume pest entomofauna, thereby highlighting their potential role as allies of forest managers and farmers.

each ASV are included in the Supporting Information S7 Table.

**Funding:** This project was supported by the Belgian National Fund for Scientific Research (FNRS; https://www.frs-fnrs.be/fr/) through a doctoral grant awarded to C.V. (FRIA-FNRS FC 36435). The funders had no role in study design, data collection and analysis, decision to publish, or preparation of the manuscript.

**Competing interests:** The authors have declared that no competing interests exist.

## Introduction

Chiroptera is the second largest mammalian order, comprising at least 1,470 species [1], 70% of which are insectivores worldwide [2] and even rises to ca. 100% in Europe [3]. Insects are, however, deeply impacted by anthropogenic alterations, with a concomitant current global decline in diversity, abundance and biomass [4]. Determining the dietary spectrum of European bats is therefore the cornerstone of their conservation, as it will enable us to evaluate the extent to which they are threatened by the plummeting insect populations but also how they possibly readjust their foraging behavior to offset this decline in their usual food source.

While insects are declining worldwide, a restricted number of arthropod pest species are thriving [4,5]. These arthropod pests can dramatically affect food production, as they are responsible for 25–50% of crop destruction [6,7]. Pest control by pesticide costs over $10 billion annually in the US [8]. Rather than relying on chemical pesticides, biological control by generalist insectivorous species could be efficient to reduce pest populations worldwide [9–13]. In fact, insectivorous bats are highly voracious controllers of potentially harmful arthropods, thereby avoiding roughly $22.9 billion/year of agricultural losses throughout North America [14–16]. A recent estimation suggested that a single breeding colony of 4,000–5,000 adult *Miniopterus schreibersii* bats preyed upon 1,610 kg of pest species over a five month period [17]. In addition, over 60 metric tons of pest arthropods were reportedly consumed daily by a subset of seven bat species across Europe [18]. Bats can thus serve both as pest controllers and 'natural samplers' of biodiversity, and could be used to unravel the presence and expansion of new arthropod pests.

However, most research on bat control of pest arthropod populations has been focused solely on crop pests [18–20], thereby largely neglecting pests detrimental to forests (but see below). Yet temperate forests—which represent over a third of Europe's surface area—provide numerous ecosystem services [21]. Previous studies have shown that bats act as effective top-down regulators of the pest pine processionary moth (*Thaumetopoea pityocampa*), by enhancing their foraging activity according to the moth abundance, thus significantly reducing the reproductive success of this pest [22]. In regard to broadleaved forests, Ancillotto [23] showcased the beneficial role of *Plecotus auritus* bats as regulators of beech forest pests, which accounted for up to 85% of their consumed prey. In addition, a recent experimental field study showed that the insect density was significantly higher on seedlings in bat-excluded plots than on control seedlings, hence highlighting the vital role of bats in structuring forest ecosystems [24]. As about 60% of European forests are threatened by climate change, with insect outbreaks accounting for 26% of potential damage [25], it is now crucial to expand our knowledge on forest pest control services provided by bats in Europe.

The regulatory action of bats has also been widely understudied in farms [16], despite the harmful effects of house flies (*Musca domestica*) and stable flies (*Stomoxys calcitrans*) on livestock [26–28]. Indeed, *S. calcitrans* commonly disturbs livestock, leading to energy loss, lower feed intake and increased stress in annoyed animals [26]. *S. calcitrans* also hampers livestock wellbeing, causing blood loss and skin lesions. Combined with their ability to transmit pathogens, fly infestations can reduce milk production by 40–60% and meat production by 25% in cows [29]. Livestock welfare regulations fostering antiparasitic treatments, insecticide applications and cowshed cleanliness may be hazardous for the preservation of insectivorous bat species due to the combined effects of loss in insect densities and secondary poisoning [30]. Yet many bat species occur in the vicinity of farms where they roost (e.g. *Myotis emarginatus* [31] and *M. mystacinus* [32]), forage (e.g. *M. nattereri*; [33]), or both (e.g. *M. emarginatus*; [34]). It is therefore urgent to gather more data regarding bat feeding on pest flies in farmland areas so as to develop bat-friendly farming strategies that optimize this behavior.

Over the course of a night, bats usually do not restrict their foraging activity to a single type of feeding habitat, alternatively visiting several of them, such as forests, meadows, orchards, hedgerows, riparian vegetation along rivers and residential areas (e.g. cowsheds; [30,35–37]). Depending on the prey availability across habitats, while being constrained by their wing morphology, insectivorous bat species display four hunting strategies [38]: aerial hawking, flycatching, trawling and gleaning. Bat species displaying various hunting strategies, i.e. showing hunting flexibility, can feed on more diverse prey and exploit heterogeneous foraging habitats, thereby ultimately decreasing their vulnerability to environmental changes [39]. Considering the impacts of climate change and the resulting ongoing landscape alterations, it is now of paramount importance to establish the links between bat foraging grounds, dietary spectra and hunting strategies, and in turn determine key landscape features that should be maintained or restored.

It is critical to take breeding season variations into account when exploring bat diets, as the dietary spectrum of reproductive females presumably changes according to the prey phenology and availability [17,40–42]. Previous studies revealed a close temporal match between bat activity, diet composition and the emergence of the main arthropod prey species [22,43,44]. Accordingly, determining how bat species with different physiological needs react to temporal variations in the arthropod prey distribution is essential to tailor appropriate conservation guidelines.

Consequently, in the light of the human-associated massive arthropod decline, we aimed to study the diet composition, its breeding season variations and pest consumption patterns of an insectivorous European bat species using Geoffroy's bat, *Myotis emarginatus* (hereafter abbreviated as "ME"), as model. We further intended to explore the hunting strategies and foraging habitats of this bat species based on the prey ecology. To achieve these goals, we monitored six ME roosting colonies in Wallonia (Belgium) at three sampling points throughout the breeding season via metabarcoding of DNA extracted from feces. This species' range extends from southern, southwestern and central Europe to Asia Minor. The study area corresponded to the northern limit of the range of this bat species where, despite a trend towards a recovery in the population size in recent decades [45], a combination of insular distribution [34] and low genetic diversity [46] makes it especially vulnerable. ME—a thermophilic species—probably encounters ecological limits in Belgium. In fact, the species is classified as "Near Threatened" on the Belgian Red List, implying that it warrants special attention [47]. It has been hypothesized that preferred prey items might be less available under suboptimal conditions at the northern margin of the geographic range of insectivorous bat species [48]. Geoffroy's bats preferentially prey on spiders, which are mostly caught by foraging in cluttered environments such as woodlands [30,31,49–52]. However, ME females have a high energetic demand during the reproductive period [53,54]. As it would be hard for these females to fulfill their increased needs by only foraging in woodlands, they appear to adopt a more opportunistic diet by feeding abundantly on cattle-related flies captured in and around cowsheds [30,31,35,52,55–57]. We thus focused on this species to overcome the current lack of studies on its diet at the northern edge of its range using the most advanced genetic tools. Indeed, only a handful of research studies have incorporated high-throughput sequencing to investigate the ME prey spectrum. Furthermore, the few cases that did so were restricted to Mediterranean [39,56] and Atlantic climatic conditions [40,56,58], while entomofauna communities are expected to widely differ under the temperate climatic conditions that prevail in Belgium.

More specifically, our first objective was to determine which arthropod orders dominated the ME diet at the northern margin of its distribution range. Here, in accordance with the findings of previous research studies carried out under temperate climatic conditions [52,55,57,59], we predicted a congruent dietary spectrum across the bat colonies, with Diptera prevailing, followed by Araneae and, to a lesser extent, Lepidoptera.

Our second objective was to explore ME breeding season variations in the dietary spectrum, as females go through gestation, parturition, lactation and post-lactation during the June to August reproduction period. Although previous studies on various other bat species outlined dietary changes in females during the breeding season (see Introduction), such variations have not been noted in ME in temperate areas of its range [52,57,59], and we therefore expected that there would be few temporal variations over the nursery season in ME diet.

Our third objective was to assess the hunting strategy and foraging habitats employed by ME, by deriving this information from ecological traits of the consumed prey. Here we hypothesized that ME would display a certain degree of flexibility by catching its prey via gleaning [39,52,55,56,59,60] and aerial hawking to a lesser degree [39,51,56,59]. We further expected that this species would take advantage of this behavioral plasticity to feed in forests, while complementing its food intake by preying on cattle flies.

Finally, our fourth objective was to assess ME pest species consumption, with a specific focus on taxa detrimental to forest trees and livestock. In this respect, ME provision of pest-control services remains unclear, with prior case studies drawing different conclusions [18,61]. However, we expected to detect stable flies (*S. calcitrans*), i.e. a cattle pest species that has been noted multiple times in ME diet samples [40,52,56,57].

## Material and methods

### A. Study sites and guano collection

During the 2021 breeding season, we sampled six *Myotis emarginatus* breeding colonies typically located in cellars or attics of old buildings in Wallonia, Belgium (S1 Fig). Only the Aubel colony was located right next to a farm building. Details on the colony selection and size are summarized in the S1 File. This bat is a strictly protected species under Annex IVa of Directive 92/43/EEC and Annex II of the Bern Convention. All localities were privately owned or protected. Access to the bat roosts was authorized by the Département de la Nature et des Forêts (DNF) authorities from the Service Public de Wallonie (SPW; permit 2021-RS-10).

Each maternity roost was visited three times at a four-week interval. Overall, samples from the first, second and third sampling sessions were collected in June, July and August, respectively (see collection dates in the S1 Table). Only the Aulne breeding colony was sampled at the very end of May, June and July, but hereafter we simply refer to these three temporal sessions as "June", "July" and "August". The sampling period encompassed distinct female breeding states. Females were generally pregnant in June and gave birth in late June-early July (C.V. personal observation). As breastfeeding lasts about four weeks in this species [62,63], it mostly occurred in July but also sometimes partly in early August depending on the roost.

We got an overview of the landscape available for ME by calculating the relative area of nine land cover classes within a 10 km radius around each breeding colony (Figs 1 and 2; see the S1 File for the explanation of the land cover classification). Classes were estimated using the LifeWatch Landcover database from 2015 [64] and QGIS tools (v3.28; QGIS Development Team). We selected this distance as it was previously reported that ME may forage up to about 10 km from their breeding colony [35,51,59]. In the study area, intensively managed grasslands consisted mostly of permanent pastures for livestock grazing and, to a lesser extent, of meadows for fodder production [65]. Wheat and corn were the dominant crops in the study area [66]. Overall, broadleaved forests in Wallonia primarily consisted of oaks, beeches, a mix of both, alongside various other temperate tree species (e.g. ash, maple, hornbeam, birch, alder and poplar [67]). Further details on the study area are outlined in the S1 File.

The samples consisted of fresh guano collected from plastic boxes that had been carefully cleansed with a DNA decontaminant (bleach 30%) and placed under the breeding colony for

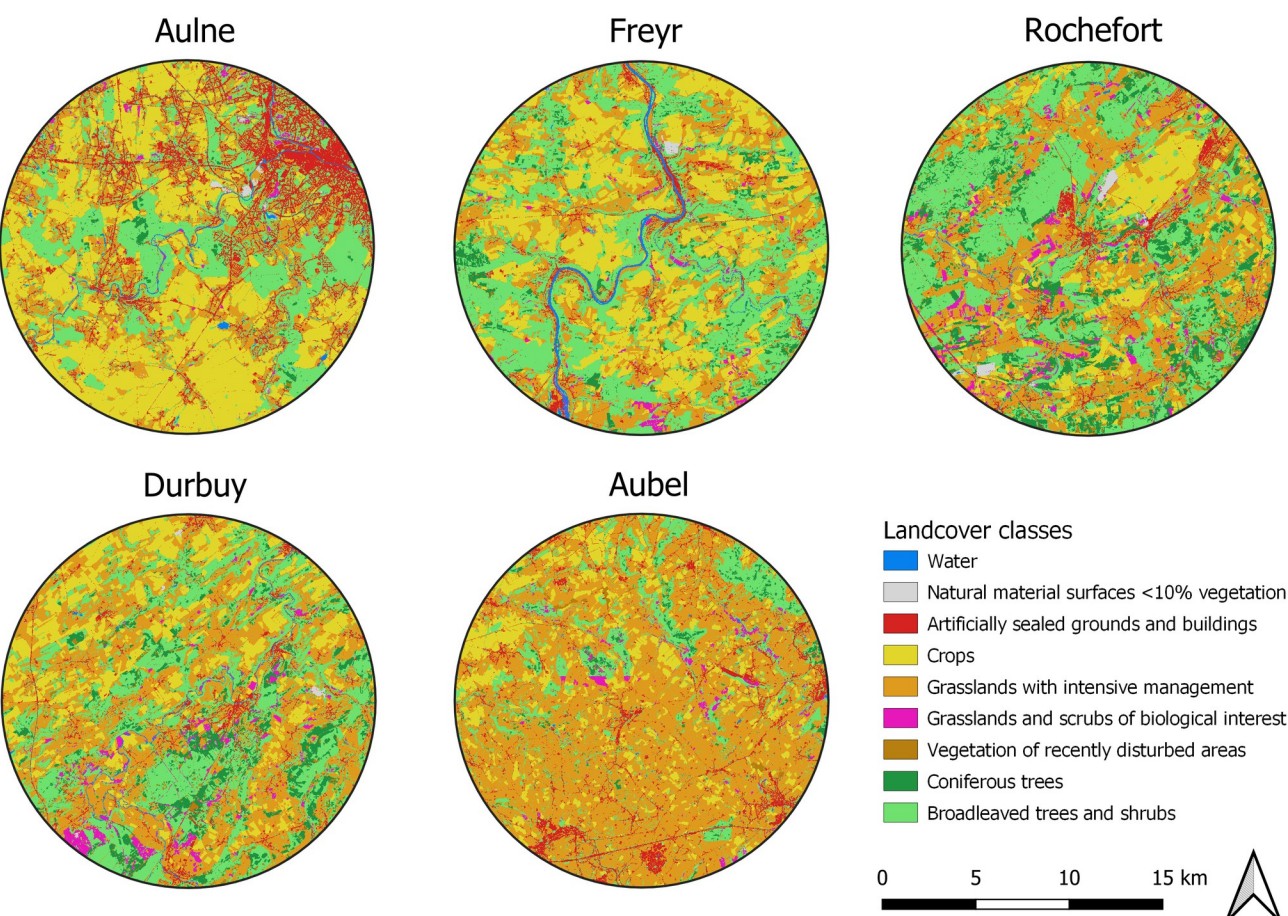

**Fig 1. Maps depicting the landscape around each maternity colony.** Nine land cover classes were displayed within a 10 km radius around the five bat colonies of *Myotis emarginatus*. Land cover classes were extracted and adapted (see the S1 File) from the 10 m resolution Landcover raster layer of 2015 created by the LifeWatch project and available at https://maps.elie.ucl.ac.be/lifewatch/ecotopes.html. Reprinted from the LifeWatch database under a CC BY license, with permission from LifeWatch Belgium, original copyright 2015.

one night (see details in the S1 File). We treated each individual fecal pellet as a distinct sample. In addition, at each maternity visit, we placed two pieces of sterilized absorbent paper (Ahlstrom-Munksjö LabSorb™, Helsinki, Finland) in the boxes to serve as field blanks in order to control for environmental contamination. Guano samples were immersed in pure ethanol and placed at -20°C within 12 hours until DNA extraction.

## B. DNA extraction, PCR amplification and sequencing

We extracted DNA from 328 samples and field blanks using a QIAamp® Fast DNA Stool Mini Kit (Qiagen, Hilden, Germany; Handbook version 02/2020), according to the manufacturer's protocol for Isolation of DNA from Stool for Pathogen Detection, with some modifications (see details in the S1 File). An extraction blank was added to each extraction batch.

We conducted a two-step PCR strategy (see details in the S1 File). In PCR$_1$, we amplified the mitochondrial *cytochrome c oxidase subunit I* (COI) using two primer pairs to decrease each primer taxonomic bias (additional reasons detailed in the S1 File). As primers, we chose both the 133 bp fragment by Galan et al. [58] (MG-LCO1490-MiSeq and modified MG-univR-MiSeq) and the 157 bp minibarcode described by Zeale et al. [68] (ZBJ-ArtF1c and ZBJ-ArtR2c). These primers are hereafter referred to as "Galan" and "Zeale".

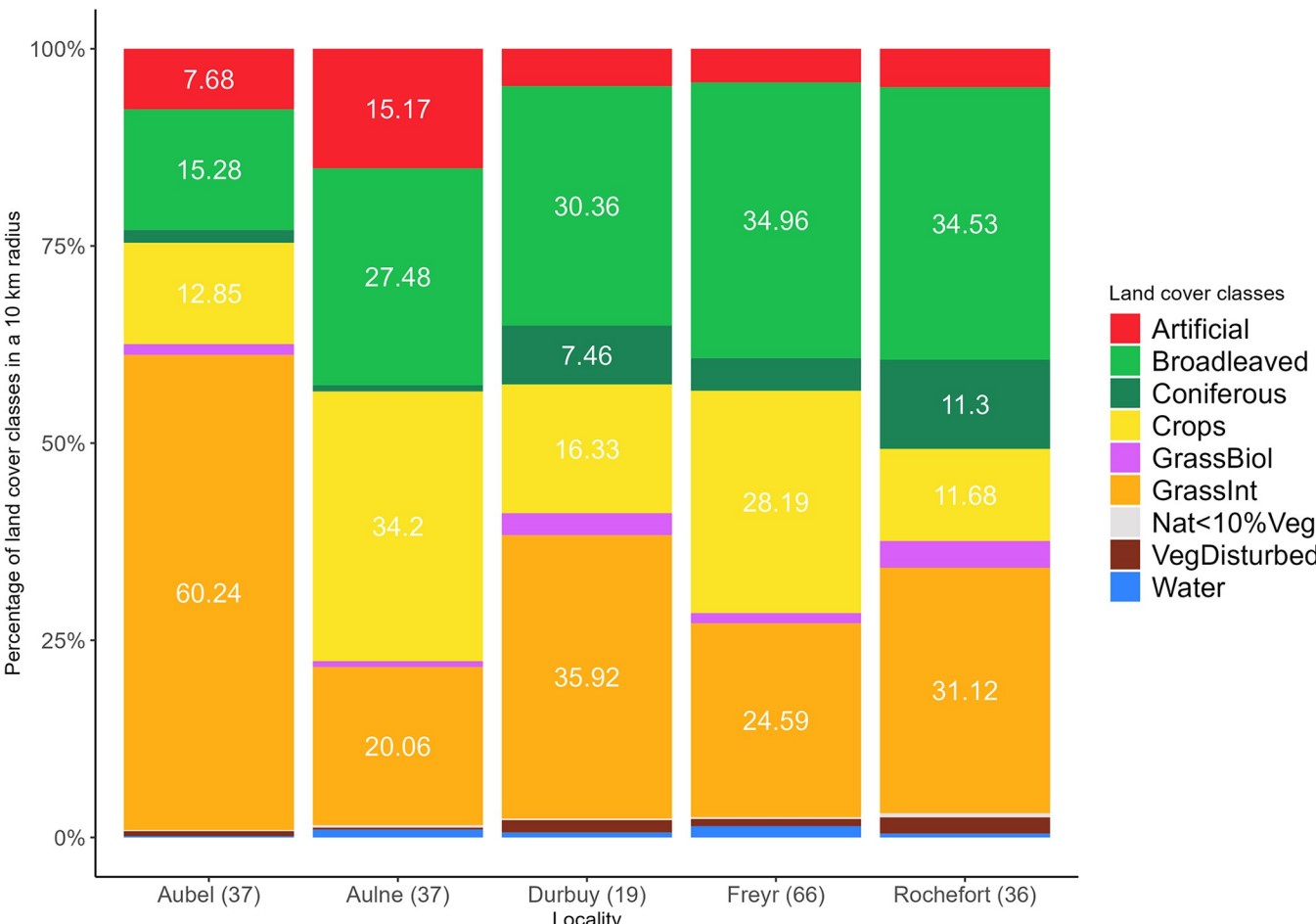

**Fig 2. Relative proportion of the land cover classes around each maternity colony.** Nine land cover classes were compiled within a 10 km radius around the five bat colonies of *Myotis emarginatus*. Land cover classes were extracted and adapted (see the S1 File) from the 10 m resolution Landcover raster layer of 2015 created by the LifeWatch project and available at https://maps.elie.ucl.ac.be/lifewatch/ecotopes.html. Percentage values above 5% were displayed on the plot. Abbreviated land cover classes: Artificial (Artificially sealed grounds & buildings); Broadleaved (Broadleaved trees); Coniferous (Coniferous trees); GrassBiol (Grasslands and scrubs of biological interest); GrassInt (Grassland with intensive management); Nat<10%Veg (Natural Material Surfaces <10% vegetation); VegDisturbed (Vegetation of recently disturbed areas). On the x-axis, the number of individual fecal pellets is between brackets.

$PCR_1$ reactions were carried out using KAPA HiFi HotStart Kits (Roche®, KAPA Biosystems, Basel, Switzerland), in a final volume of 25 μL containing 0.5 μL of 1 U/μL HotStart DNA polymerase, 0.75 μL of 10 mM dNTPs, 0.75 μL of each of the 10 μM forward and reverse primers, 5 μL of 5x buffer, 4 μL of DNA extract and 13.25 μL of molecular grade water. The thermocycler optimized conditions included an initial denaturation at 95°C for 3 min, followed by 38 cycles of denaturation at 98°C for 30 s, hybridization at 45°C for 45 s (Galan) / 52°C for 30 s (Zeale) and extension at 72°C for 30 s, and a final extension at 72°C for 5 min (Galan) / 7 min (Zeale). $PCR_1$ negative controls were included in every 96-well plate reaction.

In $PCR_2$, to trace down the sample ID (see the S1 Table) in downstream analyses, we incorporated a sample-specific combination of 8 bp long Illumina tags (i5 and i7) using the Nextera XT index kit (Illumina, San Diego, CA). $PCR_2$ was carried out in a final volume of 15 μL containing 0.3 μL of 1 U/μL HotStart DNA polymerase, 0.45 μL of 10 mM dNTPs, 1.5 μL of each of the 2 μM forward and reverse tags, 3 μL of 5x buffer, 1.5 μL of $PCR_1$ product and 6.75 μL of molecular grade water. The thermocycler optimized conditions were identical to the $PCR_1$

steps for Galan except that only 8 cycles were performed with hybridization at 55˚C for 30 s for both markers. All pre-PCR mixes were set up under a sterile hood in a DNA-free room to minimize the risk of contamination.

Following DNA purification, quantification and pooling at equimolarity (see details in the S1 File), DNA library construction and sequencing were conducted at the University of Liège GIGA Genomics platform. Finally, for each primer pair, a total of 292 samples, 36 field blanks, 31 extraction blanks and 7 $PCR_1$ blanks were sequenced on an Illumina NovaSeq flow cell.

## C. Bioinformatics and prey list construction

The sequenced libraries of Galan and Zeale COI markers yielded 84,100,577 and 73,991,168 raw paired-end reads respectively, with an average of 190,704 and 177,437 reads per sample. We controlled the quality of these demultiplexed reads with the FastQC software [69]. We then used Cutadapt [70] to remove all reads shorter than 108/104 bp (forward/reverse) and longer than 150 bp. Trimmed data was imported into QIIME 2 (v2021.8.0 [71]), where paired-end sequences were denoised, dereplicated into ASVs (Amplicon Sequence Variant) and chimeras were removed using the dada2 plugin (see parameters in the S1 File) [72].

A custom BOLD database was curated using the method described by O'Rourke with some modifications [73] (see their project's GitHub repository [74] and details in the S1 File). We trained the curated database into primer specific classifiers that were then used to assign the taxonomy of ASVs using the feature-classifier plugin in QIIME 2 (see details in the S1 File). Following taxonomy assignment, we only retained sequences with sufficient taxonomic information (at least order level) and ASVs whose assignment confidence was $\geq$ 0.98 (see details in the S1 File). Through the whole bioinformatics pipeline, we retained and identified as arthropods 6.46% (Galan) and 60.75% (Zeale) of the initial reads, which were dereplicated into a similar number of ASV features: 1,121 and 1,005.

We imported the resulting taxonomy files into the R statistical environment for subsequent analysis (v4.2.0; [75]). We minimized potential false positives and contaminations by removing, within each sample, all ASVs whose read count was lower than the read counts of the field, extraction and $PCR_1$ blanks. Whenever we could not confirm the presence of a taxonomic assignment within a 500 km radius around the study area, it was manually trimmed from the whole dataset (further exclusion criteria are detailed in the S1 File). We finally applied three filtering steps based on the read abundance (detailed in the S1 File).

Following the overall filtering process, our dataset consisted of 198 samples, containing 416 and 302 prey taxa for the Galan and Zeale primers, respectively. We investigated whether the taxa retrieval differed according to the primer pair by applying paired Wilcoxon signed-rank tests for each arthropod order using Benjamini & Hochberg adjusted p-values. For each sample, 206 prey detected by both primer pairs were merged, leaving us with a full dataset of 512 distinct biological entities (see the S2 Fig for the sample distribution per locality and session). The Galan primers enable host species identification and they revealed that only three Orval samples originated from our target species. We therefore removed this locality and associated samples from further analyses, which left us with a final dataset of 195 samples and 509 taxa. Throughout this manuscript, the "taxa" term has been used to encompass several taxonomic levels, but always with the most precise identification achievable.

One good metabarcoding practice involves the inclusion of PCR replicates, which minimize the risk of obtaining false-positives, therefore increasing the reliability of the results [39,58,76]. Yet we consciously decided not to include them, and, to reduce the risk of false-positives, we first included negative controls at every major step, we then applied a strict bioinformatic pipeline (denoising and chimera removal in dada2) and we finally filtered the data conservatively,

as detailed previously. In fact, as mentioned above, both primer pairs partly overlapped in the results, hence confirming the efficiency of the method. Finally, combining two primers targeting the COI gene without replicates has recently been recognized efficient and accurate to assess insectivorous bat diets [17,41]. Our multi-marker strategy thus allowed us to optimize the trade-off between the costs of analyses and the comprehensiveness of the results [77,78].

## D. Statistical analyses

**1. Datasets used for statistical analyses.**   We transformed raw read count data into presence/absence data (PA; S1 Table), as the former does not accurately depict the species abundance [79]. Based on the PA dataset, we calculated three different metrics that we applied on the whole dataset to fulfill the first objective (i.e. general diet composition) but also to distinct data subsets depending on the objectives: (1) frequency of taxa occurrence (FTO), which represents the number of taxa occurrences of a specific category, divided by the total number of taxa occurrences, and multiplied by 100; (2) frequency of sample occurrence (FSO), which refers to the number of samples in which a prey item was detected, divided by the total number of considered samples, and multiplied by 100 [80,81]; (3) weighted percentage of occurrence (wPO; S2 Table), a semi-quantitative index for assessing prey abundance in feces, which was calculated, for each sample, as the prey item occurrence (= 1 in PA data), divided by the total number of occurrences of all prey, and multiplied by 100 [43,81]. The metrics characteristics are detailed in the S1 File.

**2. Variations throughout the breeding season.**   We conducted the following analyses to fulfill our second objective, which was to shed light on potential dietary differentiation during the breeding season. First, to determine whether the sampling effort was sufficient for each session (i.e. sampling month), we built richness inter- and extra-polation curves of Hill numbers using the iNext package on the PA data [82,83] (see details in the S1 File).

We tested whether the prey average richness among samples differed between sessions at the order level by running generalized linear mixed models (GLMMs) with a quasi-Poisson error distribution using the glmmPQL function of the MASS package [84] (see details in the S1 File). For each arthropod order tested, the response variable was the number of taxa identified in each sample. Session was the fixed factor, while locality was introduced as a random factor to account for additional sources of variability in diet composition attributed to breeding site membership. Pairwise comparisons were computed with Tukey post-hoc tests using the glht function from the multcomp package [85].

We tested the impact of session variations on distance matrices of the ME diet with a permutational multivariate analysis of variance (PERMANOVA; see details in the S1 File) implemented in the vegan package [86]. We constructed distance matrices collapsed at two taxonomic levels, with individual fecal pellets considered as the sampling unit, using the vegan package [86]. At the species level, we calculated the Jaccard's dissimilarity index for PA data, while at the order level we built a Bray-Curtis dissimilarity index for wPO data [56]. We then performed pairwise PERMANOVA to distinguish variations between groups using the pairwiseAdonis package [87]. Alongside PERMANOVA, we conducted permutational analysis of multivariate dispersions (PERMDISP) based on 10,000 permutations to test the group variance homogeneity with the betadisper function in the vegan package [86].

We sought to detect which prey taxa contributed the most to differentiation among sessions by performing a similarity percentage analysis on the PA data (SIMPER; [88]), as implemented in the vegan package [86]. For this analysis, we only kept taxa whose contribution to the average between-group dissimilarity was ⩾ 0.01, with a p-value < 0.05.

In addition, to verify that the diet was uniform across the study area (first objective), we also ran all the above-mentioned tests based on the locality variable. Finally, we visualized separation patterns between samples from different sessions/localities with non-metric multidimensional scaling (NMDS), based on a Horn-Morisita dissimilarity matrix of wPO data, using the metaMDS function implemented in the vegan package [86] (see details in the S1 File).

**3. Ecological traits of arthropod prey species.** We fulfilled our third objective, i.e. to get an overview of ME hunting strategies and foraging habitats by conducting extensive bibliographic research, including both web portals and scientific articles, regarding the ecological requirements of the Araneae and Lepidoptera prey (see the S2 File). To get insight into Araneae ecology, we followed the spider guilds classification [89], which discriminates Araneae intro functional groups according to their hunting strategy and web structure (see classification in the S1 File).

Lepidoptera ecology was characterized by determining the life stage at which the species were caught as well as their preferred habitats. We classified Lepidoptera species into five life stage categories: "caterpillar", "imago", "both possible" when the juvenile-adult stages overlapped at the sampling date, "both sure" when the juvenile-adult stages were temporally distinct and the prey was consumed several times, and "unknown" when insufficient data was currently available.

**4. Pest species consumption.** We addressed our fourth objective, i.e. assessing the role of ME as a consumer of forest or livestock pests, by determining the pest status of all arthropod species through an in-depth literature search, as detailed in the S3 File. Our classification was comprehensive, including all taxa categorized as pest species in at least one bibliographic reference, even if the species was not considered as especially harmful locally. We classified pests according to how damaging they are (minor or major pest species) and according to the threatened group (crops, forest trees, fruit trees, livestock) or whether they are pathogen vectors or livestock pests, as described in the S1 File.

As ME has been previously reported to feed in cowsheds [30,31,49], we noted when *Bos taurus* DNA was detected in the bat fecal pellets. Cow DNA was detectable by means of secondary predation, i.e. through bats feeding on flies—mostly *S. calcitrans* and *M. domestica*—themselves feeding on cattle blood [27,56]. We examined spatiotemporal differences in the detection of these pest flies in relation to cattle by applying Kruskal-Wallis tests using the kruskal. test function from the stats package on the wPO dataset restricted to the 107 samples containing DNA from both *B. taurus* and *M. domestica* and/or *S. calcitrans*. Then, to compare pairwise differences, we conducted Dunn tests with Bonferroni adjusted p-values [90] using the rstatix package [91]. Finally, we investigated whether bat consumption of cattle-related pest flies was more frequent in areas with more cattle by calculating a Pearson correlation between the mean wPO values of cattle-related pest flies per bat colony (here again using a wPO subset with *B. taurus* DNA) and the relative number of cows located within a 10 km radius around each bat colony. Cow numbers were retrieved from Wallonia Public Service reports [65]. Through our study, p-values < 0.05 alpha threshold were considered statistically significant.

## Results

### A. General diet composition

We first aimed to describe the ME diet composition at the northern edge of its range. The 509 validated arthropod taxa belonged to three classes, 17 orders, 145 families, 357 genera and 419 species (S3 Table). The Galan primers allowed the identification of 16 orders, as compared to 14 with Zeale primers (Fig 3). In fact, all orders included more taxa using the primers from Galan, except for Lepidoptera and Trombidiformes (Fig 3). For five orders, i.e. Araneae,

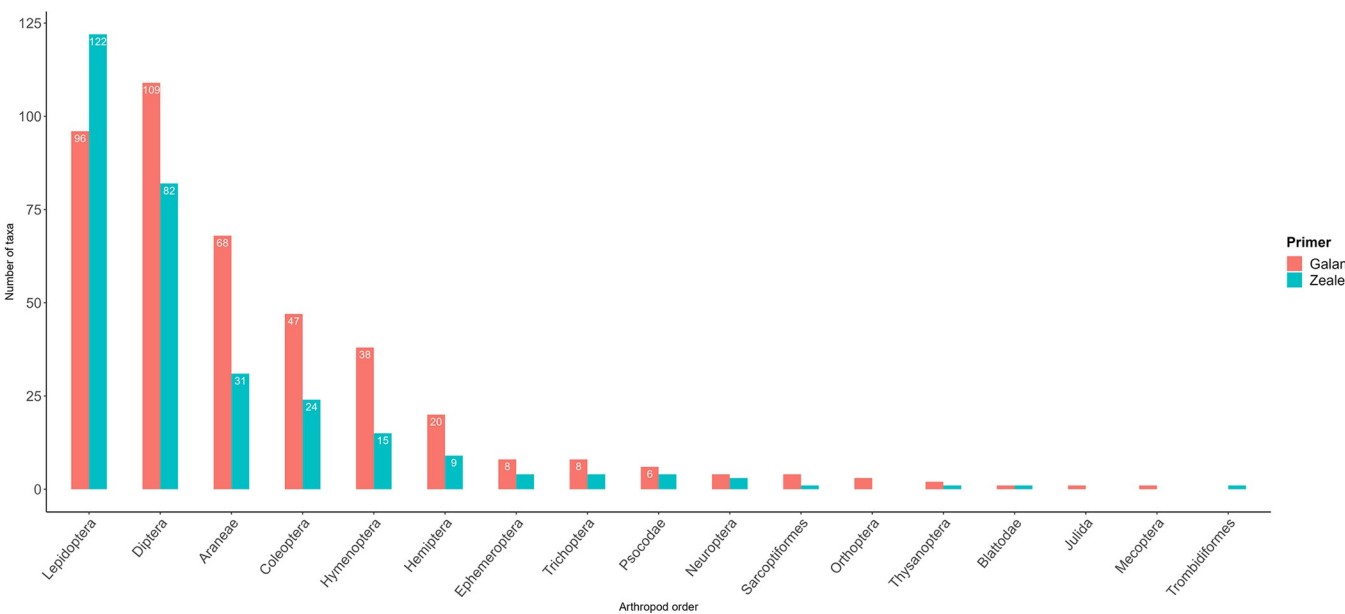

**Fig 3. Number of taxa per order.** Number of Arthropoda prey taxa recovered in the diet of *Myotis emarginatus* with the PCR primers from Galan et al. [58] and Zeale et al. [68] for the 17 taxa orders. Values above 5 were displayed on the plot.

Coleoptera, Diptera, Hymenoptera and Sarcoptiformes, the taxonomic coverage was at least twofold higher (1.3-fold for Diptera) when using Galan primers as compared to Zeale primers and these differences were significant (paired Wilcoxon signed-rank tests: all V > 1514, all p < 0.001).

The taxonomic resolution was high as we identified 82.32% of the taxa to the species level, 11.39% to the genus level and only 6.29% to the family or order level. Around 77% of the prey taxa were assigned to four main orders (Fig 3): Diptera (26.52%), Lepidoptera (26.33%), Araneae (13.95%) and Coleoptera (10.81%). Among the remaining orders, Hymenoptera—mainly Ichneumonidae, Hemiptera, Trichoptera, Ephemeroptera and Psocodae each represented between 1–10% of the prey taxa diversity, while far behind, eight orders each represented < 1% of the dietary spectrum (Fig 3). In terms of frequency of taxa occurrence (FTO), Araneae ranked first, as 27.62% of overall taxa occurrences involved this order, closely followed by Lepidoptera (27.09%) and Diptera (25.03%). The remaining 14 orders each accounted for less than 7% FTO. In terms of the frequency of sample occurrences (FSO), Diptera, Lepidoptera and Araneae were detected in 98.97%, 83.59% and 76.41% of samples, respectively.

Each sample contained a mean of 14.43 taxa (range: 1–49). Each taxon was on average detected in 5.53 samples (range: 1–133), but 41.45% taxa occurred in a single sample, while six taxa were recorded in > 50 samples: the Diptera species *Stomoxys calcitrans* (n = 133) and *Musca domestica* (n = 126), the Lepidoptera species *Agriopis marginaria* (n = 64) and *A. leucophaearia* (n = 54) and the Araneae species *Nuctenea umbratica* (n = 77) and *Cyclosa conica* (n = 50). In the overall dietary dataset, only 43 taxa (8.45%) and 65 taxa (12.77%) were retrieved from the five remaining localities and the three sessions, respectively.

We performed additional analyses with the "locality" variable as part of the first objective to evaluate whether diet was congruent across the study area. These generally pointed in the same direction, i.e. few differences in diet diversity among the colonies at the species level (accumulation curves: Fig 1 in the S4 File) and at the order level (GLMM: Fig 2 and Table 1 in

**Table 1. GLMM outputs of the sessions pairwise comparisons.**

| Fixed effect | Estimate | Std.Error | z-value | P-level | Stat. Difference | Order |
|---|---|---|---|---|---|---|
| July-June | -1.265 | 0.161 | -7.858 | <1e-05 | July < June | Araneae |
| August-June | -1.226 | 0.153 | -8.026 | <1e-05 | August < June | Araneae |
| August-July | 0.039 | 0.196 | 0.198 | 0.978 | ns | Araneae |
| July-June | -0.359 | 0.114 | -3.161 | 0.004 | July < June | Diptera |
| August-June | -0.157 | 0.105 | -1.499 | 0.291 | ns | Diptera |
| August-July | 0.202 | 0.117 | 1.72 | 0.195 | ns | Diptera |
| July-June | -0.771 | 0.133 | -5.821 | < 0.001 | July < June | Lepidoptera |
| August-June | -1.330 | 0.158 | -8.418 | < 0.001 | August < June | Lepidoptera |
| August-July | -0.558 | 0.179 | -3.124 | 0.005 | August < July | Lepidoptera |
| July-June | -0.913 | 0.191 | -4.785 | < 1e-04 | July < June | Coleoptera |
| August-June | -2.334 | 0.333 | -6.999 | < 1e-04 | August < June | Coleoptera |
| August-July | -1.422 | 0.357 | -3.978 | < 0.001 | August < July | Coleoptera |
| July-June | -1.795 | 0.240 | -7.466 | < 0.001 | July < June | Hymenoptera |
| August-June | -2.721 | 0.350 | -7.766 | < 0.001 | August < June | Hymenoptera |
| August-July | -0.926 | 0.406 | -2.279 | 0.055 | ns | Hymenoptera |
| July-June | -1.655 | 0.407 | -4.062 | < 0.001 | July < June | Hemiptera |
| August-June | -1.533 | 0.371 | -4.129 | < 0.001 | August < June | Hemiptera |
| August-July | 0.122 | 0.504 | 0.242 | 0.968 | ns | Hemiptera |

Parameter estimates (quasi-Poisson distribution) of the sessions pairwise comparisons calculated from the generalized linear mixed models (glmmPQL function; equation Order ~ session + (1 | locality)). All effects are shown. Std.Error = Standard error; ns = not significant.

the S4 File), while we noted differences in diet composition between the colonies, especially at the species level (PERMANOVA: Table 2 in the S4 File and SIMPER: S4 File and S4 Table).

## B. Variations throughout the breeding season

We secondly aimed to investigate the diversity and compositional variations in the ME diet throughout the June to August breeding season. None of the richness extrapolation curves reached a plateau for any of the sessions (Fig 4). These curves suggested that between 60% (July) and 63% (August) of the potential prey richness was recovered. In addition, the visualizations indicated that the richness diversity of prey taxa was higher in June as compared to both July and August (Fig 4), and these differences were significant as the confidence intervals (95%) just partly overlapped between the latter two.

GLMMs revealed that the prey richness at the order level was at least twofold higher in June as compared to July and August for Araneae, Lepidoptera, Coleoptera, Hymenoptera and Hemiptera, and these differences were significant (Fig 5, Table 1). For Diptera, there was only a significantly higher prey richness in June as compared to July and the change factor was < 2 (Fig 5B, Table 1). For Lepidoptera and Coleoptera, the models also indicated that the prey richness at the order level was higher in July compared to August (Fig 5C and 5D, Table 1).

PERMANOVA tests were used to assess the dissimilarity between groups of samples. This test accounts for both within-group and between-group variability to determine the significance of the observed differences. At the species level, PERMANOVA multivariate tests revealed a different taxonomic composition over the sampling sessions (df = 2, $R^2$ = 0.102, F = 10.911, p < 0.001), and the pairwise comparisons were all significant (Table 2). PERMDISP tests, which account for within-group variability to assess the group variance homogeneity, were non-significant for the session variable (df = 2, F = 0.446, p = 0.634). At the species

**Table 2. PERMANOVA outputs of the sessions pairwise comparisons.**

| Pairwise comparison | F-statistic (F) | $R^2$ | Degrees of freedom (df) | p-value (p) |
|---|---|---|---|---|
| **At the species level** | | | | |
| June-July | F = 13.096 | $R^2$ = 0.094 | df = 1 | p = 0.001 |
| June-August | F = 16.776 | $R^2$ = 0.114 | df = 1 | p = 0.001 |
| July-August | F = 2.790 | $R^2$ = 0.021 | df = 1 | p = 0.001 |
| **At the order level** | | | | |
| June-July | F = 27.758 | $R^2$ = 0.181 | df = 1 | p = 0.001 |
| June-August | F = 48.619 | $R^2$ = 0.271 | df = 1 | p = 0.001 |
| July-August | F = 4.973 | $R^2$ = 0.038 | df = 1 | p = 0.007 |

Multivariate tests outputs of sessions pairwise comparisons of Myotis emarginatus diet composition at the species level (using Jaccard's dissimilarity index for presence absence data) and at the order level (using Bray-Curtis dissimilarity index for wPO–weighted percentage of occurrence data: Within each sample, it is the prey item occurrence/total number of occurrences of all prey*100).

level, session was the most discriminating factor as 10.21% ($R^2$ = 0.1021) of the variation in diet composition was explained by its group membership, compared to only 3.61% ($R^2$ = 0.0361) for the locality (S4 File). At the order level, compositional differences remained significant over the sessions (df = 2, $R^2$ = 0.202, F = 24.355, p < 0.001), and the pairwise comparisons were still all significant (Table 2). However, PERMDISP tests showed that dissimilarities

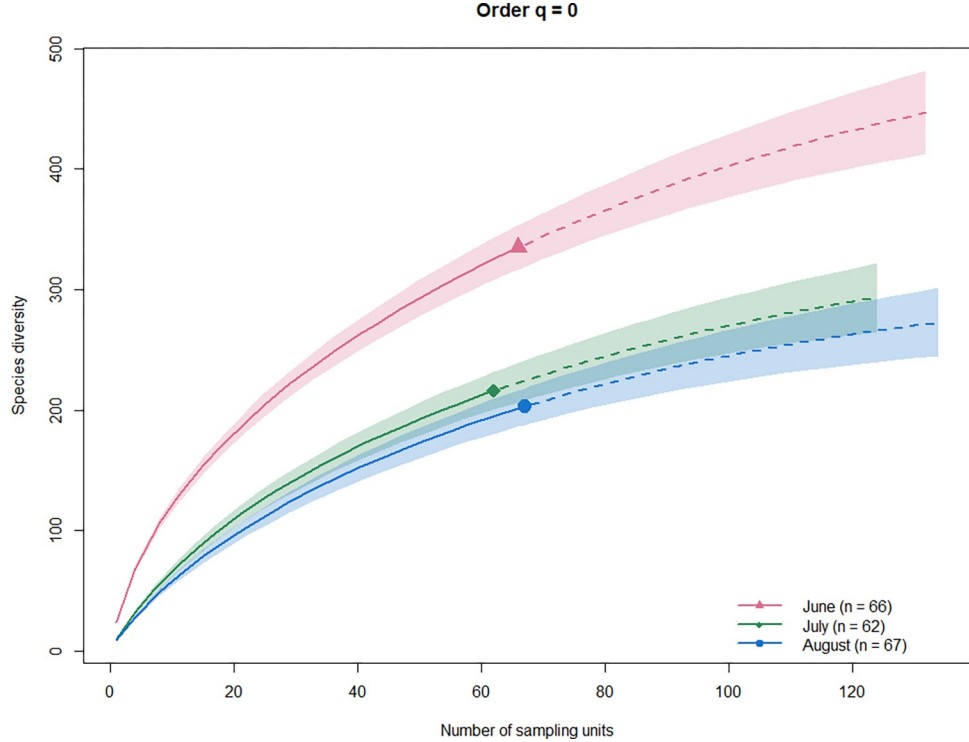

**Fig 4. Accumulation curves of taxa species diversity.** Alpha diversity was based on Hill numbers calculated for q = 0, corresponding to prey richness. Curves are drawn for each sampling session and according to the number of guano bat samples analyzed for their diet. Shape symbols represent the observed values and dashed lines the extrapolated values expected with increased sampling effort. Shaded areas represent 95% confidence intervals. The observed percentage of sampled richness (observed value/estimated value*100) was: 60.95% (June), 60.08% (July) and 62.79% (August). In the legend, n = number of individual fecal pellets.

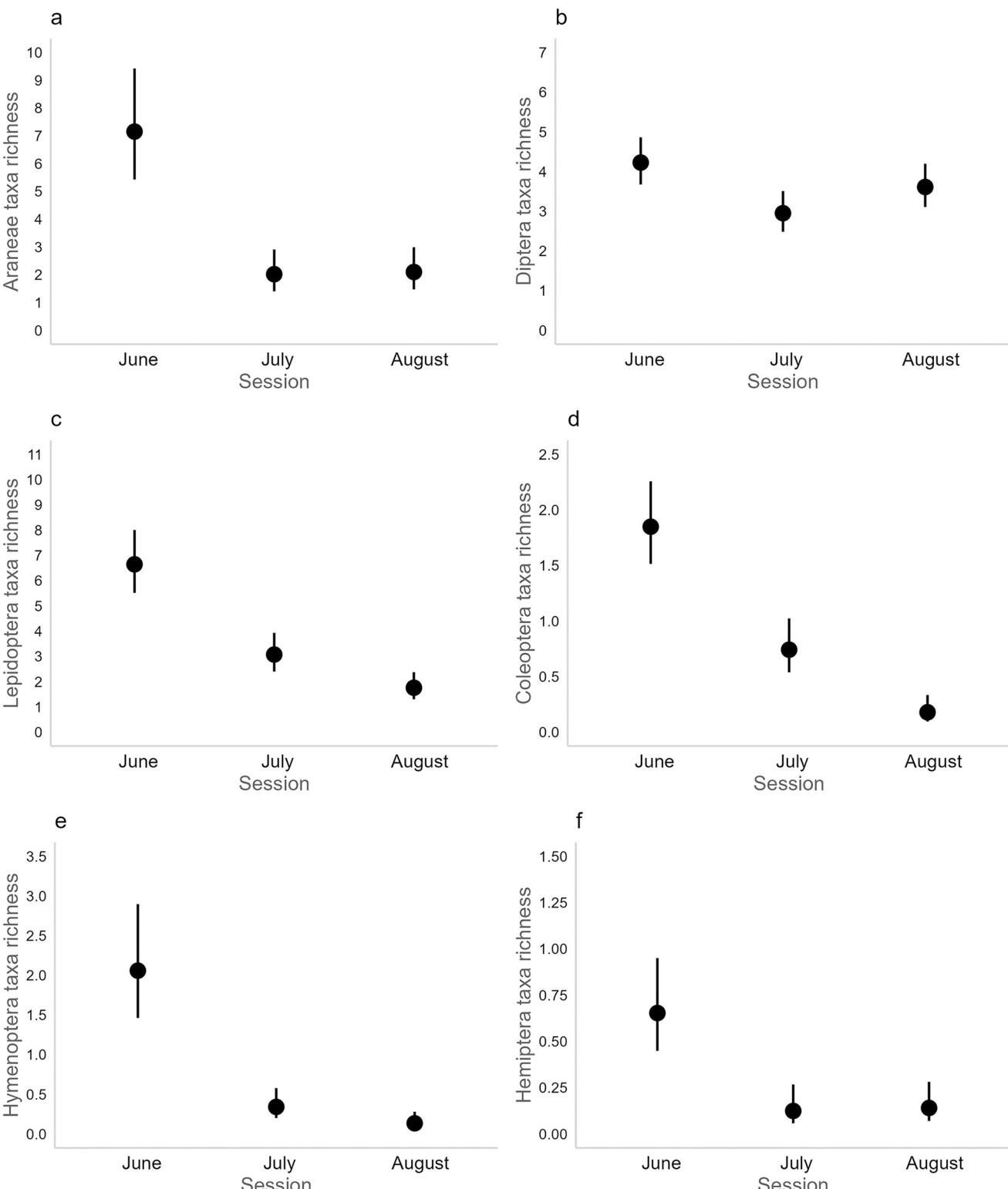

**Fig 5. GLMM temporal visual outputs.** Probability estimates (with 95% confidence intervals) of the taxa richness eaten by *Myotis emarginatus* per sampling session (June, July and August) and for the most consumed orders, as inferred by generalized linear mixed models with a quasi-Poisson distribution (glmmPQL function; equation: Order ~ session + (1 | locality)).

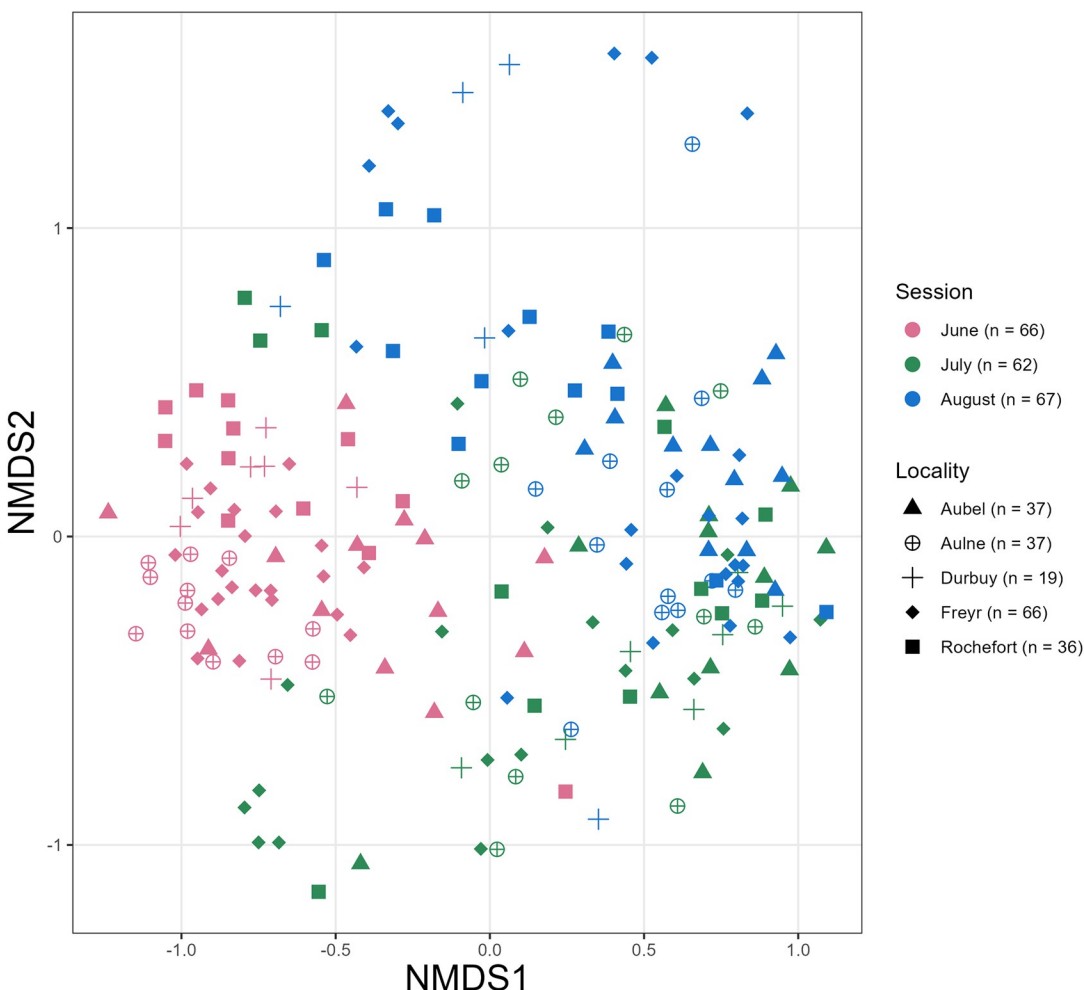

**Fig 6. NMDS visual output.** Non-Metric Multidimensional Scaling, based on a Horn-Morisita dissimilarity matrix constructed on the weighted percentage of occurrence data of the diet (wPO: Within each sample, it is the prey item occurrence/total number of occurrences of all prey*100). Colors indicate the sampling session: June, July and August. Shape symbols represent the sampling location of the *Myotis emarginatus'* droppings in Wallonia, Belgium: Aubel, Aulne, Durbuy, Freyr and Rochefort. In the legend, n = number of individual fecal pellets.

between sessions could have been driven by differences in dispersion of the samples within each session (df = 2, F = 26.895, p < 0.001).

Overall, the diet composition dissimilarity between individual fecal pellets averaged 82.3% according to the SIMPER analysis. The main drivers of differentiation were attributed to feeding on *Agriopis marginaria*, *A. leucophaearia* and *Nuctenea umbratica* between June and both July and August, while they were related to the consumption of *Limonia nubeculosa*, *Musca domestica* and *Zeiraphera isertana* between July and August (S5 Table).

In regard to overall variations in diet composition, the NMDS plot suggested that locality was not the main driver of differentiation, as inferred by the great overlap in the shape points (Fig 6). However, the color points revealed temporal changes in prey composition, with individual fecal pellets from June showing little mixing with those from July and August (Fig 6). The compositional differences were therefore likely spatially minor but more prominent temporally, with a shift in prey composition at the end of June, and this pattern was in line with the niche overlap values (see Morisita-Horn index calculation in the S5 File).

## C. Ecological traits of arthropod prey species

**1. Araneae.**   As third objective, we aimed to use life history traits from prey species to infer ME behaviors. Overall, 70 Araneae taxa belonged to 12 spider guilds (S3 Fig). Among this variety of functional groups, 60.1% of the taxa occurrences (FTO) concerned the orb-weaver spider guild, consisting of three distinct families: Araneidae (48.06%), Tetragnathidae (10.36%) and Uloboridae (1.68%). Then tangle-web spiders from the Theridiidae family represented 14.9% FTO. Sac spiders belonging to the Clubionidae (7.38%) and Anyphaenidae families (5.31%) together accounted for 12.69% FTO. The nine other spider guilds each represented < 4% FTO, but included several ambusher and stalker guilds, such as cellar, crab and zebra spiders.

**2. Lepidoptera.**   Overall, 134 Lepidoptera taxa were detected in 163 individual fecal pellets. While only 24 taxa were presumably eaten as caterpillars, this concerned 82 individual fecal pellets and accounted for 34.91% of the Lepidoptera FTO. Concerning the prey likely consumed as imagos, this involved 45 taxa, which were retrieved from 101 individual fecal pellets but only accounted for 19.03% FTO. As many Lepidoptera juvenile-adult stages temporally overlap, distinction was not possible in 47 species, accounting for 29.53% FTO (corresponding to the "both possible" category). Results revealed that caterpillar consumption occurred in every bat colony, with related wPO values ranging from 15 to 31.9% (Fig 7).

Regarding the habitat occupied by Lepidoptera species, 81 taxa (~68%) frequently used broadleaved forests, whereas coniferous trees were visited by only ~10% of the Lepidoptera taxa. Gardens, hedgerows and scrubs were the other preferred habitats, each being used by ≥ 35% of the consumed Lepidoptera taxa, according to the bibliographic research findings.

## D. Pest species consumption

As fourth objective, we investigated the ME potential as pest consumer, with a specific focus on forest and livestock pests. 96.92% of the individual fecal pellets contained at least one of the 58 identified potential pest taxa (see full list in the S6 Table). Pest species accounted for 24.03% of all taxa occurrences (FTO). While 44 pest species belonged to the Lepidoptera, the two most consumed pests were the Diptera species *S. calcitrans* and *M. domestica*, which were found in 68.21% and 64.62% of the samples (FSO), respectively. Then the broadly polyphagous Lepidoptera species *Agriopis marginaria*, *A. leucophaearia* and *A. aurantiaria* were retrieved from 32.82%, 27.69% and 13.33% of the samples (FSO), respectively. Finally, Lepidoptera oak feeders *Tortrix viridana*, *Apocheima hispidaria* and *Orthosia cruda* were detected in 10–15% of the samples (FSO). Among the 56 pests assigned to the species level, 30.36% had major effects on their host. Regarding the type of targeted host, 73.20% of the pest species fed on trees, while only 16.10% could potentially damage crops. Finally, 10.70% of the pest species were potential vector pathogens for either livestock (the Diptera species *Eristalis tenax*, *Culicoides punctatus* and *M. domestica*)—one of which also presenting a zoonotic risk (*S. calcitrans*)—or plants (the Coleoptera species *Ips typographus* and the Hemiptera species *Rhopalosiphum padi*).

To address the spatiotemporal differences in the consumption of livestock pests, we built a dataset of 107 samples, including DNA of both cattle fly pests and cow. Among these, 102, 99 and 2 samples contained *S. calcitrans*, *M. domestica* and *Culex* sp. DNA, respectively. *S. calcitrans* and *M. domestica* consumption did not differ significantly between the localities, as revealed by the Kruskal-Wallis tests (H = 6.068, df = 4, p = 0.194 & H = 7.051, df = 4, p = 0.133). Consumption of both *S. calcitrans* and *M. domestica* did however vary over the breeding season (H = 15.069, df = 2, p < 0.001 & H = 9.271, df = 2, p = 0.001), with each species being much less preyed upon in June as compared to July and August (*S. calcitrans*: June-July pairwise p = 0.009; June-August pairwise p < 0.001 & *M. domestica*: June-July pairwise

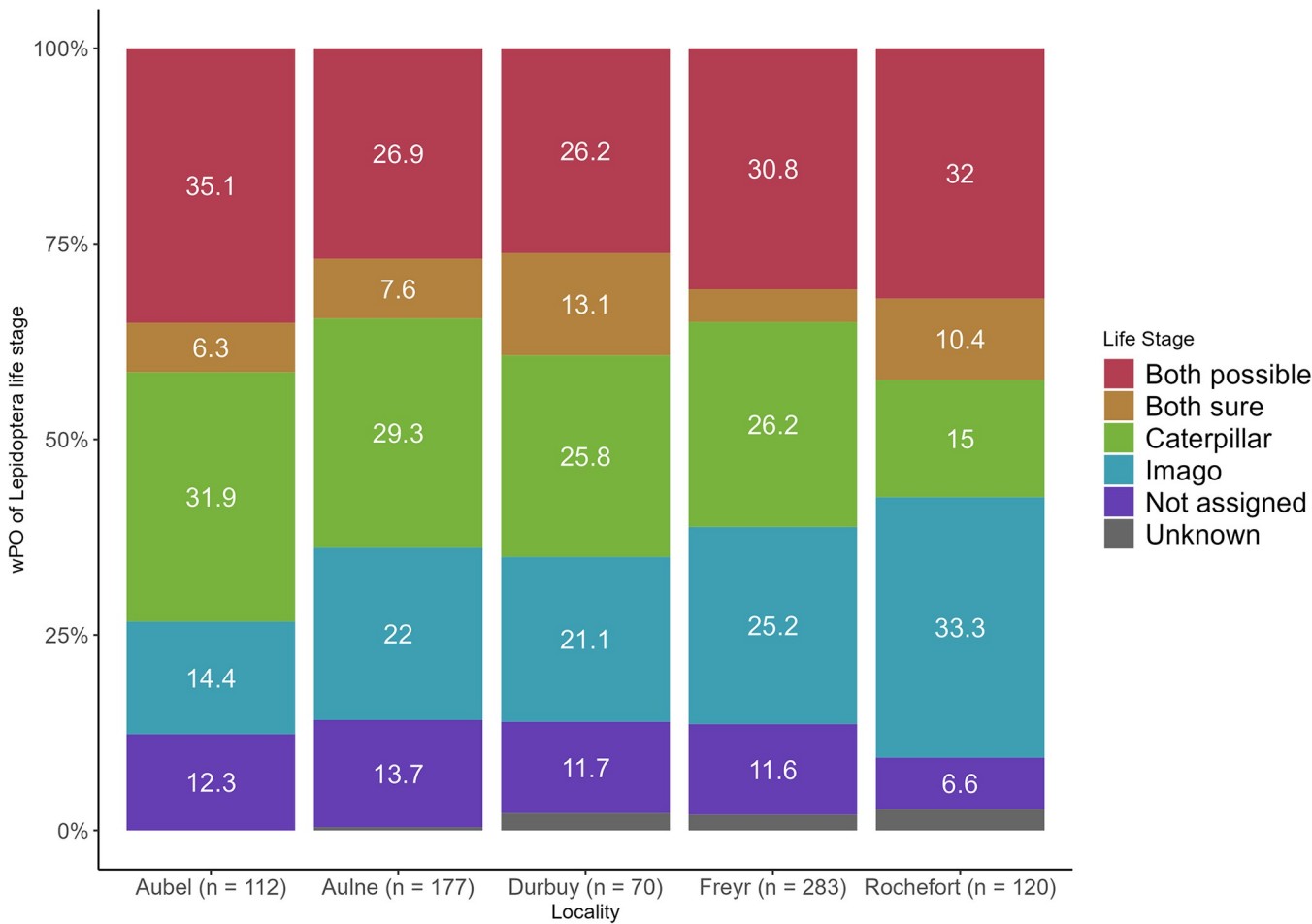

**Fig 7. Lepidoptera life stage.** Weighted percentage of occurrence (wPO: Within each sample, it is the prey item occurrence/total number of occurrences of all prey*100) of the Lepidoptera taxa according to their life stage in the diet of each sampled colony of *Myotis emarginatus*. wPO values above 5% were displayed on the plot. Life stage: Both possible (distinction between imago and caterpillar is not possible as the two stages temporally overlap at the time of sampling), Both sure (taxa retrieved both during the caterpillar stage and imago stage for sure), Caterpillar, Imago, Not assigned (because taxa not detected at the species level), Unknown (currently not enough data on the life stage). On the x-axis, n = number of taxa occurrences.

p = 0.043; June-August pairwise p = 0.007). Across all localities and sessions, 55.38% (FSO) of the individual fecal pellets contained DNA of either or both *S. calcitrans* or *M. domestica* and the related bats were attested to forage around cattle. However, this number rose to 75.97% FSO when restricted to July and August. Finally, there was no correlation between the proportion of eaten cattle fly pests (that had been attested to feed on cows) contributing to the ME diet and the number of cows within a 10 km radius around the colony ($r^2 = 0.221$, p = 0.721; Fig 8).

## Discussion

This is the first study carried out using advanced genetic tools to investigate variations in the breeding season diet and pest consumption of the European bat *Myotis emarginatus* at the northern limit of its range. Here we revealed that this bat species is a generalist that fed on a wide array of Diptera, Araneae and Lepidoptera taxa. Overall, we detected potential pest taxa in about 97% of the samples. The beginning of the breeding season (June) was marked by a diet rich in orb-weaver spiders living in cluttered vegetation, in accordance with its archetype diet. However, during the parturition and lactation period (July and August)—an extremely

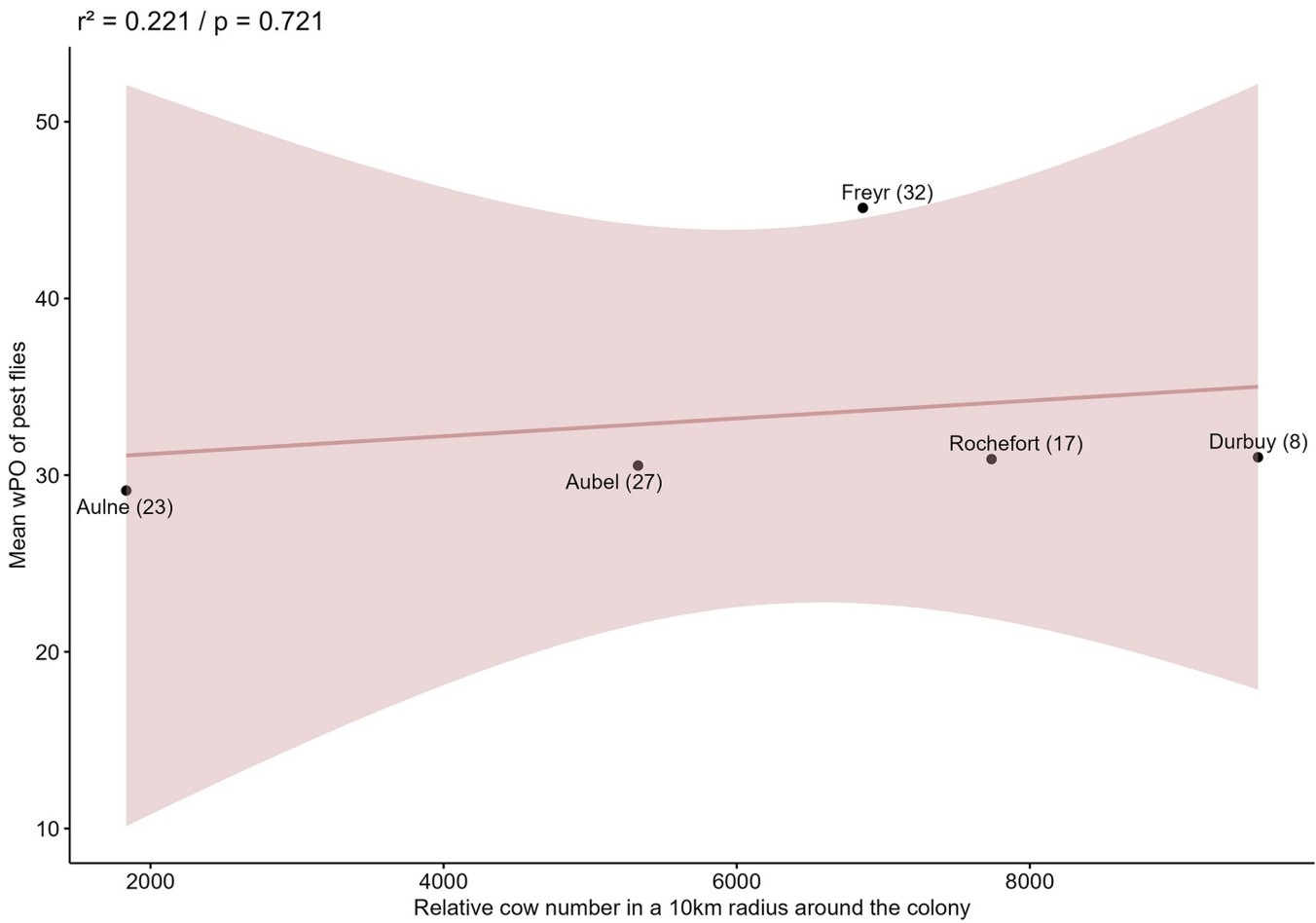

**Fig 8. Correlation between pest flies and cows' numbers.** Pearson correlation between the mean weighted percentage of occurrence (wPO: Within each sample, it is the prey item occurrence/total number of occurrences of all prey*100) values of pest flies attested to feed on cattle (*Stomoxys calcitrans*, *Musca domestica*, *Culex* sp.) eaten by *Myotis emarginatus* in each colony and the relative cow (either meat cattle or dairy cattle) numbers within a 10 km radius around each sampled colony. The number of individual fecal pellets considered by colony is shown between brackets. The shaded area represents the 95% confidence interval. The correlation is positive but weak ($r^2 = 0.221$) and not significant (p = 0.721).

energy demanding time for females—we suggest that the latter maximized their food intake by feeding on cattle fly pests, which are hyperabundant around cows [59]. In August, freshly weaned young bats might also have benefited from this easy-to-catch and plentiful fly food resource. We therefore speculate that ME is able to thrive in the northern part of its range depending on the availability of several landscape features, including broadleaved forests and cattle farms.

## A. Using two primer pairs enables extensive prey retrieval

The diet of insectivorous bat species such as ME can be thoroughly described using a dual-primer metabarcoding approach. Indeed, in 195 samples, 17 Arthropoda orders were retrieved (Fig 3), which is greater than reported in previous metabarcoding studies on the ME diet in southern Europe over a similar [56] or even longer time frame [39,40] compared to that considered here. As such, the combination of two COI minibarcodes enhanced the taxonomic coverage, while also reducing the risk of false-negative results [77,92,93]. In fact, Zeale et al. [68] described non-degenerated primers, which, despite being conventionally used to highlight the trophic niche of insectivorous bats, are hampered by several amplification biases attributed

to primer mismatches [78,94]. Consequently, they failed to detect several orders, such as Orthoptera and Mecoptera (Insecta class) and Julida (Diplopoda class), which we identified for the first time in the ME diet using the Galan primers (Fig 3). These were efficient in detecting a wide range of orders (n = 16) due to their high level of degeneracy (Fig 3). The discrepancy in taxa retrieval between the primer pairs was especially acute for some core-food orders, such as Araneae and Coleoptera, which had a higher detection level with Galan than with Zeale primers. Surprisingly, even the number of Diptera taxa was significantly higher using Galan primers, while Zeale primers were previously considered efficient in retrieving both Diptera and Lepidoptera taxa in bat diets [43,93,95]. This might have been due to the increased use of the efficient Galan adapted primer pair in recent studies of invertebrate and vertebrate diets [76,77,96–100]. The benefit of Galan primers was also their capacity to amplify host DNA, enabling simultaneous identification of the predator and its prey.

Although the rarefaction curves suggested that about 40–50% of the prey species were under-detected depending on the roosting site (Fig 1 in the S4 File), it is not that surprising seeing that we only sampled each colony three times in a single year. Moreover, this range was in line with that noted in previous metabarcoding diet studies [101–104]. As such, we are convinced that our sampling design was the most cost-efficient in assessing the ME diet at the Wallonia regional scale, as we detected an unprecedented number of prey at the species level (> 80%), combined with the fact that each individual fecal pellet displayed a high richness of prey taxa (mean: 14.43 taxa). Moreover, we retrieved a broad range of potentially new species in Belgium (S3 Table). Overall, only six taxa were detected across all maternity roosts and in > 50 of the samples, while more than 40% of the taxa occurred in a single sample. These results are in line with findings on *R. ferrumequinum*, which harbors a core diet composed of a handful of prey taxa common to all colonies, combined to a secondary diet composed of many rare taxa specific to breeding colonies and collection dates [105].

Note that secondary predation—the prey of ME prey—potentially inflated our richness estimations. Indeed, DNA from ME meals was indistinguishable from DNA traces from the gut content of its prey, which is a well-known bias in metabarcoding studies [81,106]. In ME, concern may arise from the order of Araneae, because most of them are carnivorous and prey upon other arthropods, which may also be eaten by ME. This complicates the distinction between primary and secondary predation. However, secondary prey are unlikely to be substantially detected as their DNA will have been degraded by two gut passages, leaving only fragmentary traces barely amplified by PCR [61]. We nevertheless sought to offset this issue by applying strict bioinformatic filters to remove taxa with a low read count (see the S1 File for details). Moreover, there was congruent identification for half and two-thirds of the prey taxa identified via Galan and Zeale primers, respectively, thereby boosting our data reliability. In fact, secondary predation might be helpful to identify unexpected trophic interactions and feeding habitats [56,58,61], although these findings should be cautiously interpreted as we cannot completely eliminate field contamination. For instance, the detection of *B. taurus* DNA in bat samples containing cattle-related flies suggested that these bats caught their prey in the vicinity of cattle. Overall, ME is thus an excellent natural sampler of biodiversity at the regional scale. Non-invasive sampling in its colonies is hence relevant to detect rare and new species, to unveil trophic interactions, and—if carried on over several years—to monitor alterations in insect populations due to anthropogenic pressure.

## B. *M. emarginatus* eats spiders, flies and caterpillars

Fitting our first hypothesis, the bulk of the ME diet consisted of three orders: Araneae, Diptera and Lepidoptera. ME is one of the only three species known to have an Araneae-based diet

worldwide [107,108]. As such, this taxon is a major food source for ME bats, constituting up to 70% [57,60] or even up to an astonishing 80% [51] in pellet volume in previous studies, whereas it ranged from 11 to 80% in relative frequency in metabarcoding studies [39,40,56]. In fact, a spider-rich diet has been proposed as the diet archetype closest to this bat's natural condition [51]. Regarding the Diptera prevalence, previous research also detected this order in > 90% of samples in Germany and Switzerland [55,57].

While ME have long been known to feed on Araneae and Diptera species [40,52,56,57,59], to our knowledge this is the first time that Lepidoptera has been recognized as being such an important food resource for ME bats at the northern border of their range. Indeed, whereas Lepidoptera consumption ranged from 15 to 20% in relative frequency in southern Europe [39,56], it did not account for more than 6% of the ME diet by volume in the northwestern part of the continent [52,57]. This discrepancy could probably be explained by the fact that the meridional estimates were based on molecular techniques, while the continental studies used a conventional method, with prey remaining in bat droppings being visually identified by their morphological traits. In fact, pieces of Lepidoptera species are generally difficult to identify [109], as the most easily recognizable parts of the imago body (i.e. the wings) are usually bitten off before ingestion [59]. In addition, soft-bodied caterpillars are almost entirely digested, thereby leading to underestimation of this order when using morphological methods. Yet in our study caterpillars seemed to account for a relevant percentage of occurrence (wPO) of the Lepidoptera prey ingested at each locality (Fig 7), while also representing around 35% of the taxa occurrences (FTO). Moreover, the two most consumed Lepidoptera prey (*Agriopis marginaria* and *A. leucophaearia*) were both presumably preyed upon as larvae. Our results provided support for the recent suggestion that another gleaner species (*Plecotus auritus*) potentially prey on caterpillars [23]. To our knowledge, a single study recently investigated whether European bats feed on imagos versus larvae [110]. Among the 129 Lepidoptera species they identified, 23 (~18%) of them had been eaten in caterpillar form, which is in agreement with the 24 species (~18%) reported here. This highlights the importance of using the most advanced genetic tools to accurately assess insectivorous diets.

## C. *M. emarginatus* is a generalist forager

We expected to find few variations in the ME diet in the study area, which spanned a relatively small spatial scale (116 km maximum between sites) as compared to the species' range. As predicted in our first hypothesis, we noted no large intercolonial distinction in prey composition at the northernmost edge of ME range, especially at the order level (Table 1 and 2 in the S4 File).

Yet the dietary spectrum was not uniform across the breeding sites at the species level as < 10% of the prey taxa were eaten by bats from all five breeding colonies, while less frequent prey items accounted for the largest share of the diet (41.45%). In addition, the diet composition dissimilarity between samples exceeded 80%, while the richness diversity was similar at all roosting sites (Fig 1 in the S4 File). This feeding differentiation pattern corroborated the findings of a previous telemetry study on ME that revealed clear individual differences in habitat occupation [30], with each location offering distinct arthropod communities. As mentioned in Alberdi et al. [39], a generalist species can be a composite assemblage of specialist individuals. Despite its specialized high reliance on Araneae prey, ME can be considered as a generalist forager, feeding on available arthropods encountered, albeit focusing on an array of preferred arthropod orders.

## D. Shift from a spider-rich to a fly-rich diet

Contrary to our second hypothesis, we identified a marked discrepancy and little overlap in the ME diet between June and July-August (Fig 6, Table 2, S5 File). Overall, the taxonomic differences could be explained by the higher richness diversity in arthropod prey at the onset of the nursery season at both the species (Fig 4) and order levels for five taxa (Araneae, Lepidoptera, Coleoptera, Hymenoptera and Hemiptera; Fig 5). This decrease in diet diversity from spring to summer has been noted in a variety of insectivorous bat species (e.g. [111,112]), and notably in ME in southern Europe [40]. Beyond phenological shifts in entomofauna availability, this period-based feeding behavior could also suggest that bats are more selective in their food choices following parturition [113].

Most of the seasonal changes in diet were attributed to two orders, i.e. Araneae and Diptera. In June, Araneae prey contributed the most to the ME diet, representing ~30% wPO (S4 Fig), presumably because Araneae constitute the natural original food core of ME [51,60]. Moreover, this food source is providential in late spring as it is directly available when bats emerge from hibernation, when there is lower insect abundance. In accordance with our observation, Araneae also reportedly dominated the dietary spectrum in samples from spring [56] / June [40] in southern Europe. We then found that Diptera became a predominant food source, accounting for ~40% wPO in July and even ~50% wPO in August (S4 Fig). More specifically, we propose that this dominance was driven by the great reliance on stable (*S. calcitrans*) and house (*M. domestica*) flies since, among the 88 Diptera species preyed upon by ME in July and August, these two flies accounted for around 50% of the taxa occurrences, and we noted this in every colony. ME bats could gradually shift their diet in response to transitory peaks in cattle fly availability, since Diptera upsurges are triggered by hot summer peaks but we currently cannot fully verify this hypothesis as we did not measure prey availability. Nonetheless, temporal shifts in preferred prey have been observed in several insectivorous bat species (e.g. [17,40]). These results are also in line with those reported by Andriollo et al. [43], who found that in summertime *Plecotus* bats switched their foraging strategy to target a few temporarily abundant prey. Overall, it was previously observed that bat activity during the breeding season generally depended on the insect abundance [10,114] and that foraging behavior was driven by valuable prey availability [115,116].

Importantly, we detected cattle DNA alongside these flies' DNA in 75.97% of the July and August samples, suggesting that there was substantial foraging in cowsheds and their vicinity, in line with our expectations and with prior research. Indeed, in northwestern and central Europe, ME individuals from colonies located close to farming landscapes preferentially fed on cattle-associated flies in the vicinity of cattle barns [35,52,56,57,59]. Bat feeding on cattle flies was also highly suspected in southern Europe as *S. calcitrans* has been detected in 96.4% and 68.0% of the guano samples from colonies surrounded by cattle farming activities [56]. Vallejo et al. observed a trend similar to ours, i.e. a diet dominated by Araneae prey in spring and by cattle flies in summer [56]. However, in a recent study by the same team, *S. calcitrans* consumption peaked in May and gradually declined over time, while the proportion of Araneae increased until September [40]. According to the authors, the opposite fly-spider temporal trends observed by the two studies could notably result from the distinct foraging habitat opportunities in the Iberian Peninsula, with a predominance of farming landscapes versus broadleaved forests and coniferous plantations in the first and second study areas, respectively [40,56]. In the latter, bats were therefore restricted to their primal prey source, i.e. Araneae. Our results resembled those from the former study [56], probably because the landscape composition was more alike, i.e. containing large cattle farming areas, compared to the second study [40]. Indeed, cattle grazing pastures occurred in 20–60% of the potential flight zone

depending on the bat maternity colony in this study (grasslands managed intensively in the Figs 1 and 2).

The observed shift in food resources was concurrent with juveniles' birth (C.V. personal observation). As parturition and lactation involve high energy expenditure for females [53,54], this could explain their tendency to rely on the local superabundance of flies in the vicinity of cowsheds (up to 10–40 house flies/m$^2$ of ceiling [59]). A recent study similarly related temporal dietary differences with energy requirements of females during the lactation period [117]. In addition, as weaning of young starts in August [62], such a great reliance on cattle flies could also be attributed to the presence of inexperienced young accompanying adults in search for easy-to-catch and abundant food.

Feeding on *S. calcitrans* and *M. domestica* was not influenced by the number of cattle within a 10 km radius around the colony, although secondary predation on cows was observed (Fig 8 (S4 Table)). Note also that none of the locality pairwise comparisons of the diet (S4 Table) included these two flies, thus implying that their consumption was not a driver of differentiation between sites, but instead that these taxa were preyed upon in each colony (Kruskal-Wallis tests). We therefore revealed that feeding on cattle flies was not restricted to the colonies located in agricultural landscapes (e.g. Aubel) but occurred in all nursery roosts, regardless of whether livestock were nearby. In accordance with our results, Brinkmann (unpublished data) found that eight radio-tracked female ME bats flew further to feed in cowsheds instead of foraging in available forest stands near their colony in southern Germany. Hence, feeding around cattle would not be as opportunistic and punctual as previously thought for this bat species [51,52,56,59,118], and we instead suggest that ME bats may actively search for cowsheds and surrounding pastures, which then become privileged foraging sites during the breeding period, as they offer an extremely abundant and reliable food source in the suboptimal conditions of the northern margin of ME range [35]. Here bats would thus temporally switch their otherwise spider-rich to a fly-rich diet, thereby highlighting their foraging behavior plasticity.

### E. *M. emarginatus* is an agile gleaner forest forager

In line with our third hypothesis, ME consumed a broad range of non-flying prey species, such as Lepidoptera caterpillars and Araneae (Figs 7 and S3), confirming that this species mainly feeds by substrate gleaning. This was already reported by previous studies with regard to ME foraging strategies, as inferred from analyses of echolocation signals during foraging [49], telemetry data [59], wing morphometric data [38] or diet data [39,52,55,56]. Yet gleaning is not mandatory, as bats could prey upon flies in cowsheds by gleaning them off ceilings or walls, or otherwise via aerial hawking, as described by Krull et al. [59].

Further corroborating our third hypothesis, when feeding in natural settings, this bat species would mostly forage in broadleaved forests, which represented about a third of the landscape around each colony, except for the Aubel colony where they only accounted for 15% of the landscape (Figs 1 and 2). Indeed, the most caught spiders belonged to the orb-weaver guild (S3 Fig), which thrive in high vertical vegetation structure as it offers more anchoring points for their webs [119]. ME thus seemed to principally forage in closed and dense habitats such as forests or shrubby hedges, which is supported by its echolocation calls and wing morphology [49,59]. In addition, about 70% of the Lepidoptera species appeared to occur near broadleaved trees, with nearly a quarter of them relying heavily on *Quercus* spp. Our finding that ME would occupy oak woodlands is in accordance with the results of a previous study carried out in the Netherlands [30]. Finally, while substantial exploitation of coniferous plantations by meridional bat colonies has been reported [50,51,56], these foraging grounds seemed frequently avoided in Belgium, as only 10% of the Lepidoptera species relied on conifers,

probably because few native coniferous trees stand were available around the study roosts (Figs 1 and 2).

We finally identified several ambusher or stalker spider guilds, which lack a prey-capture web but instead display an active predation style involving chasing down and subduing their prey (S3 Fig). The fact that these guilds are usually highly mobile and fast predators indicates that ME bats displays a very agile flight with good maneuvering skills.

Although ME foraging habitat and behavior were inferred from the ecology of their prey, we are confident that this species likely exploited these habitats for feeding. Indeed, they are corroborated by the results of previous research involving radio-tracked ME females in temperate climatic conditions [30,31,59]. Furthermore, prey trait-based studies are increasingly being carried out and are relevant for investigating insectivorous bat diets [17,41,43,117,120].

ME bats are therefore able to exploit a wide variety of hunting grounds, while combining several foraging strategies, which highlights its high adaptability and flexibility, thereby further supporting its generalist predatory character described above [49]. Moreover, this plasticity also makes it less vulnerable to environmental changes [39], and thus less likely to be affected by drops in insect biomass and diversity or by phenological shifts induced by global warming and related phenomena (e.g. increased temperatures and altered precipitation patterns [5]).

## F. *M. emarginatus* is an ally of broadleaved forest managers and farmers

Regarding our fourth objective, we found that ME bats preyed upon a wide array of pest species (> 50 taxa) that may have minor (39 taxa) and major (17 taxa) impacts on various hosts (S6 Table). These estimated numbers were far above those reported in a previous study where only six pest species were identified in 35 samples [61]. Here we detected a high and low number of forest (41) and crop (9) pest species, respectively, which was in line with the habitat use of this bat species. Indeed, most crops are grown in an open farmland habitat, which is unsuitable for ME foraging [50].

Among the tree pest species detected, 10 may have a major effect on their host plant, mostly due to folivore larvae outbreaks, but also due to potential pathogen vectors (*Ips typographus* and *Rhopalosiphum padi*). Most major defoliators were polyphagous Lepidoptera species, affecting multiple broadleaved trees and shrubs species, including fruit trees/shrubs (e.g. *Malus*, *Prunus*, *Pyrus* and *Rubus* genera). Yet several Lepidoptera taxa fed preferentially on oaks, upon which they may have disastrous impacts. These notably included *Agriopis leucophaearia*, *Apocheima hispidaria*, *Orthosia cruda*, *Erannis defoliaria* and *Tortrix viridana* taxa, which we detected in 54, 24, 22, 17 and 26 samples, respectively. Note that all of these were exclusively consumed as caterpillars, except *T. viridana* whose life stage was uncertain at the sampling date (category "both possible"). By targeting oak-defoliating insects, ME could be especially beneficial for forest health. In fact, a recent experimental field study showed that folivore insects reduced the leaf area of *Quercus* spp. by ninefold in bat-excluded plots as compared to control forest plots [24]. Defoliated trees could be hampered by reduced growth and competitive ability, and higher levels of insect-associated pathogen infections [121]. Here, in line with previous research [10,23,24], we therefore suggest that broadleaved forests greatly benefit from the presence of insectivorous bats.

Among the six pest species being potential pathogen vectors or livestock pests, two Diptera species that are closely associated with cattle farming, namely *S. calcitrans* and *M. domestica*, were the most frequently captured. Besides its direct nuisances (see Introduction), *S. calcitrans* is generally considered to be a mechanical disease vector, potentially inoculating susceptible hosts when moving from an infected animal to another during its blood meal (reviewed in [26,27]). House flies may also be detrimental to livestock, as some bacteria like *Escherichia coli*,

*Shigella* spp. and *Salmonella* spp. Have been isolated from their external surface and digestive tract [122]. In addition, experiments have shown that they could transmit *E. coli* to cattle [28]. Note that we retrieved *Culicoides punctatus* in one sample, a biting midge involved in blue-tongue virus transmission, which is particularly harmful for cattle and sheep in northern Europe [123].

Chemical and technical (e.g. electrocution grids) solutions have been widely used to control parasitic flies in livestock [26,27], but bats might represent an alternative efficient biological method. Indeed, our findings indicated that ME bats could act as a top-down suppressor of cattle pest flies. As bats visit the same foraging sites on consecutive nights [30,59], conservation initiatives should be geared towards identifying pastures/sheds that are exploited by ME bats to ensure that the foraging conditions remain suitable for this species.

## Conclusion

Overall, to preserve near-threatened *Myotis emarginatus* bats at the northern border of their range, broadleaved woodlands hosting *Quercus* spp., cowsheds and surrounding pastures, hedgerows, shrubs, scrubs and insects-friendly gardens and the appropriate connection of these features in the landscape should be the focus of bat conservation strategies.

This study revealed that ME has developed into a synanthropic bat species, as is also the case with many other European bats [16]. Indeed, this spider-specialized bat species is able to tap the opportunity offered by the all-you-can-eat fly buffet around cattle when the energy requirements of these bats are high.

Chiroptera taxa display diverse feeding strategies, with insectivory by far being the most widespread. Yet our results highlighted a previously overlooked phenomenon regarding temperate insectivorous bats, i.e. their economic and ecological value in terms of livestock welfare and forest protection.

## Supporting information

**S1 Fig. Sampling sites.** Location of the six sampled maternity colonies of *Myotis emarginatus* studied for their diet in Wallonia, Belgium.
(TIF)

**S2 Fig. Number of samples analyzed per sampling session and breeding colony (locality).**
(TIF)

**S3 Fig. Spider guilds.** weighted percentage of occurrence (wPO: Within each sample, it is the prey item occurrence/total number of occurrences of all prey*100) of the Araneae taxa according to their spider guilds (based on the classification developed by Uetz et al. in 1999 [89]), in the diet of each sampled colony of *Myotis emarginatus*. wPO values above 5% were displayed on the plot. In the orb-weaver spiders, Ara, Tet and Ulo stands for the families Araneidae, Tetragnathidae and Uloboridae, respectively. Abbreviations of the spider guilds are detailed in the S1 File. On the x-axis, n = number of taxa occurrences.
(TIF)

**S4 Fig. wPO at the order level.** weighted percentage of occurrence (wPO: Within each sample, it is the prey item occurrence/total number of occurrences of all prey*100) of the prey taxa eaten by *Myotis emarginatus* according to their taxonomic order and to the sampling session (June, July, S3). wPO values above 5% were displayed on the plot. On the x-axis, n = number of individual fecal pellets.
(TIF)

**S1 Table. Presence absence data of the 509 taxa across the five localities.** Column abbreviations: ME (*Myotis emarginatus*); Rf (*Rhinolophus ferrumequinum*); CD (Collection Date expressed in days counted from the first collection day); ED (Extraction Date expressed in days counted from the first extraction day); NumberME (Number of adults and juveniles *M. emarginatus*); ColonySize (Number of adults and juveniles *M. emarginatus* and *R. ferrumequinum*).
(XLSX)

**S2 Table. wPO data of the 509 taxa across the five localities.** wPO (weighted percentage of occurrence): Within each sample, it is the prey item occurrence/total number of occurrences of all prey*100. Column abbreviations: ME (*Myotis emarginatus*); Rf (*Rhinolophus ferrumequinum*); CD (Collection Date expressed in days counted from the first collection day); ED (Extraction Date expressed in days counted from the first extraction day); NumberME (Number of adults and juveniles *M. emarginatus*); ColonySize (Number of adults and juveniles *M. emarginatus* and *R. ferrumequinum*).
(XLSX)

**S3 Table. Taxa number of occurrences, presence/absence in the study area and status.** The legend of the column names and color code are detailed in the second sheet of the file.
(XLSX)

**S4 Table. Simper pairwise outputs for localities.** The legend of the column names and color code are detailed in the last sheet of the file.
(XLSX)

**S5 Table. Simper pairwise outputs for sessions.** The legend of the column names and color code are detailed in the last sheet of the file.
(XLSX)

**S6 Table. List of pest species.** Number of occurrences of the pest species detected in the diet of *Myotis emarginatus*, whether they are major/minor pests and their target host.
(XLSX)

**S7 Table. List of taxa along with their fasta sequence and confidence of assignation.**
(XLSX)

**S1 File. Detailed material and methods.**
(PDF)

**S2 File. List of references used to determine the Araneae hunting strategy and web structure and the Lepidoptera habitat use and life cycle.**
(PDF)

**S3 File. List of references used to determine the pest status of Arthropoda prey.**
(PDF)

**S4 File. Results of the variations in diet according to the locality variable.**
(PDF)

**S5 File. Niche overlap calculation using the Morisita-Horn index.**
(PDF)

## Acknowledgments

The authors are thankful to the DNF (Département de la Nature et des Forêts) for allowing access to the bat roosts to apply our experimental design. We would also like to warmly thank the private owners of buildings hosting these bat colonies for welcoming us. We are especially grateful for any volunteer who devoted time and energy to the sampling. In particular, this work would not have been possible without the technical and methodological support of the Plecotus division of Natagora. We sincerely thank them for their implication in this project. We are also grateful to Thomas Duchesne for his kind help regarding cartographic analyses. Research into Lepidoptera life stages has emerged from interesting discussions with Damien Gailly; many thanks to him for his precious advices and ideas. We finally thank all members of the GeCoLAB for their fruitful conversations regarding the laboratory and bioinformatics processes.

## Author Contributions

**Conceptualization:** Chloé Vescera, Cécile Van Vyve, Quentin Smits, Johan R. Michaux.

**Data curation:** Chloé Vescera.

**Formal analysis:** Chloé Vescera.

**Funding acquisition:** Chloé Vescera, Johan R. Michaux.

**Investigation:** Chloé Vescera.

**Methodology:** Chloé Vescera, Cécile Van Vyve, Quentin Smits.

**Project administration:** Johan R. Michaux.

**Resources:** Johan R. Michaux.

**Supervision:** Johan R. Michaux.

**Validation:** Chloé Vescera, Johan R. Michaux.

**Visualization:** Chloé Vescera.

**Writing – original draft:** Chloé Vescera.

**Writing – review & editing:** Chloé Vescera, Cécile Van Vyve, Quentin Smits, Johan R. Michaux.

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
