## [Decision Letter · Decision Letter 0]

26 Dec 2023

PONE-D-23-33429All-you-can-eat buffet: A spider-specialized bat species (*Myotis emarginatus*) turns into a pest flies’ eater in cowshedsPLOS ONE

Dear Dr. Vescera,

Thank you for submitting your manuscript to PLOS ONE. After careful consideration, we feel that it has merit but does not fully meet PLOS ONE’s publication criteria as it currently stands. Therefore, we invite you to submit a revised version of the manuscript that addresses the points raised during the review process.

We look forward to receiving your revised manuscript.

Kind regards,

Camille Lebarbenchon

Academic Editor

PLOS ONE

Journal Requirements:

3. We note that you have referenced (Brinkmann, unpublished) on page 29, which has currently not yet been accepted for publication. Please remove this from your References and amend this to state in the body of your manuscript: (ie “Bewick et al. [Unpublished]”) as detailed online in our guide for authors

4. We note that [Figure 1] in your submission contain [map/satellite] images which may be copyrighted. All PLOS content is published under the Creative Commons Attribution License (CC BY 4.0), which means that the manuscript, images, and Supporting Information files will be freely available online, and any third party is permitted to access, download, copy, distribute, and use these materials in any way, even commercially, with proper attribution. For these reasons, we cannot publish previously copyrighted maps or satellite images created using proprietary data, such as Google software (Google Maps, Street View, and Earth). For more information, see our copyright guidelines: http://journals.plos.org/plosone/s/licenses-and-copyright.

Reviewers' comments:

Reviewer's Responses to Questions

**Comments to the Author**

1. Is the manuscript technically sound, and do the data support the conclusions?

Reviewer #1: Partly

Reviewer #2: No

Reviewer #3: Partly

2. Has the statistical analysis been performed appropriately and rigorously? 

Reviewer #1: N/A

Reviewer #2: No

Reviewer #3: Yes

3. Have the authors made all data underlying the findings in their manuscript fully available?

Reviewer #1: No

Reviewer #2: Yes

Reviewer #3: Yes

4. Is the manuscript presented in an intelligible fashion and written in standard English?

Reviewer #1: Yes

Reviewer #2: No

Reviewer #3: Yes

5. Review Comments to the Author

Reviewer #1: Summary:

This study investigates the diet of the Geoffroy bat (Myotis emarginatus, ME) in several sites at different times during the breeding season. Fecal pellets were collected beneath roosting sites and analyzed with two COI metabarcoding that were further compared based on their respective detection success. The authors had several objectives: exploring the spatial and temporal variation of the diet and assessing the hunting strategy and foraging habitat used by ME by looking at prey ecological trait and habitat. The obtained results were also used to determine if the processed samples were a sufficient representation of the total Geoffroy bat diet.

General Comments:

I enjoyed this interesting work. The necessity to study the diet of Geoffroy bats in combination with the landscape is well explained. The methods are overall well described (there is some missing information, and some stats should be clarified), and most interpretations are supported by the results (in my opinion a few claims should be toned down in the Discussion). I also suggest including a section on secondary predation (e.g. it preys on spiders that prey on insects and/or other arthropods) in the discussion as it could have a big impact on diet richness estimation. Generally, I found the manuscript lengthy and sometimes hard to follow, so I believe the manuscript would benefit from being condensed a little bit (filler and unnecessary words could also be removed, e.g. of unnecessary words: “as a matter of fact”).

Note: I haven’t been able to access PRJNA1019357 (it was not found in SRA).

L46: You highlighted that ME prey on some pest species and could potentially be considered a pest regulator. We can’t say from your data whether they have an impact on pest population as we just know that the species were detected in the feces.

L165-169: This section should not be in Introduction.

Note: There is an arbitrary writing rule on using numerals for numbers 10 and greater and spelling out numbers one through nine.

L193: Why did you chose to refer to S1, S2, S3? I would recommend keeping the month instead of sessions (June/July/August or gestation/lactation/post-lactation). That would be so helpful to understand your data (linking your observations to the bat life cycle, energetic needs etc).

Figure 1: It is a pretty map, but as your paper focuses on landscape (forest, cowsheds) it would be more useful for us to know the landscape surrounding the sampling sites (forest, farm land etc.: CORINE land cover map?) instead of Wallonia regions.

L205: How long were they placed there? Indicate which decontaminant (e.g. bleach 20%).

L206: What do you mean by proximity? A few cm, m? Do we have any information in the literature regarding ME individual location in their breeding sites? Do they stay to the same spot all the time? What about when individuals (adult-adult, adult-offspring/young) stay very close to each others? I believe ME can also make clusters. I’m not convinced by your assumption of 1 pellet = 1 individual and I believe this should be modified in the entire manuscript.

L229: This is indeed uncommon as using technical replicates is a golden standard. The fact that you used two markers is clearly not enough to justify the absence of replicates. It is expected that the two primer pairs won’t give the same results (that’s why you combine two markers) so you cannot use the absence of overlap between them to detect false positive (+ they have their own biases). You must rather emphasize here how you managed to decrease false positive detections using bioinformatics (denoising, chimera removal, etc) and further filtering (negative controls, min number of reads etc) in R/Excel. You filtering approaches must be conservative to compensate for the lack of technical replicability.

L232/241: Indicate the concentration of the Hot start polymerase.

L251: Indicate here or in the S1 File the percentage of PhiX that was used for the sequencing run.

L265: And the coverage percentage? If the identity is 100% but the coverage is only 5%, then the taxonomic assignment is not good.

L266: It’s always amazing to see how good the Galan’s primers are, with such a low % assigned to arthropods it still has a greater species coverage than arthropod specific primers like Zeale. Neat!

L274: Did you keep strictly ME samples or also samples with a mix of both species? If the latter please justify here as it would have big consequences on your diet assessment.

L275: How did you deal with redundancy or pseudogenes in your dataset? For example, when you both detected a taxa at the species and genus or family level in the same sample?

L303: Not sure to get the sentence. You established or aimed at establishing whether B. taurus DNA was present in ME samples through secondary predation?

L309: Indicate here that species richness is equivalent to q = 0 for Hill numbers (q = 0 is also indicated in Figure 3b but not 3a)

L310: “Working on sample coverage”? Please clarify what you did.

L313: I’m surprised that there wasn’t any overdispersion of the data, have you checked?

L315: Not clear in which case locality was a fixed effect, when session was a fixed effect and why it wasn’t always the same variable as fixed/random effect. Could you clarify this section please?

L316: individual bats or individual pellet? See comment above on the assumption 1 pellet = 1 individual.

L320: Have you checked the assumption of the PERMANOVA? It would be valuable to run a betadisper() alongside PERMANOVA, because a significant output might mean that homogeneity/heterogeneity of variances contribute to any observed differences in PERMANOVA

L324: You mean for further analyses (IBD, NMDS)?

L331: Indicate how many try and trymax in metaMDS. Please also indicate the stress value on your NMDS figures as it indicates the quality of the solution: usually a stress values lower than 0.05 indicate that the solution is of excellent quality and stress values higher than 0.2 indicate solution of poor quality (Kruskal, 1964).

L334: closed landscapes: do you mean forests or also other types of closed landscape?

L343: Replace “determined” by “estimated”

L348: “We investigated the potential link between locality and caterpillar life stage with Kruskal-Wallis analyses” I don’t understand what you tested here: for example whether there more caterpillars at locality A than B? One test per category? (corrected for multiple test?)

L359: I’m a little bit confused again with the stats here. Is Chi² good to use here? Why these two species get another test and on another dataset?

L364: Could other domestic (e.g. horses, sheep, donkey) and wild (e.g. deers) animals be important for flies?

Figure2: text is too small / resolution not good enough. A tip: you can use “theme(axis.text.x = element_text(angle = 50, vjust = 1, hjust = 1))” to rotate a bit your arthropod name so they fit entirely (no need to shorten them).

L391: Very interesting, did you also look at taxonomic resolution between the two markers? Is Galan or Zeale more resolutive than the other?

L402: you can remove “different”.

L403: Very cool to see this pattern in ME too (most taxa are rare, i.e. occurring in one sample --- similar results were observed for the greater horseshoe bat ; Tournayre et al. 2021. eDNA metabarcoding reveals a core and secondary diets of the greater horseshoe bat with strong spatio-temporal plasticity)

L409: In this section you include some interpretations that don’t belong to a Results section (e.g. “highlighting few unsampled taxa at a temporal scale while sampling effort could have been improved spatially.”

L435: if you use a function/package to retrieve and compare the contrast (Table 1, Fig 4), please indicate it in the Methods.

L469: you can put the betadisper results here

Page 19-20: formatting is weird here, probably when everything has been merged into a single pdf.

L489, 558: I’m not convinced by “individual bat” (see previous comments)

L535: Please, tone down a little bit, for example by using “presumably” (very likely though not known for certain) eaten as caterpillar, imago etc

L553: Again it would be nice to see on your map the different forests to make the link with you’re saying here.

L573 from S1, S2,and S3, respectively (respectively is usually at the end in english)

L576: did not differ at all?

L608: There has been another study published last Spring: See Vallejo et al. 2023 they studied five maternity colonies in the Basque Country every 2 weeks for the whole duration of the maternity season. Like you they found significant changes in diet diversity and composition but the opposite way “At the beginning of the season, M. emarginatus consumed a variety of prey orders, Diptera being the most abundant; but as the season progressed, the relative consumption of Araneae increased.” I highly recommend you reading this paper to nourish your discussion.

L637: The Galan and Zeale target fragments highly overlap so I doubt the reason is new sequences in databases. I rather think that Zeale used to be the most commonly used in insectivorous diet studies so most people were sticking with it without trying to get better, and now you and others are showing that minibarcodes with degenerate primers such as Galan’s can do much better. It’s interesting because this questions what we use as standards and how that has biased our knowledge on bat diet.

L640-644: I agree: it amplifies the predator DNA and yet it still detects more arthropod than Zeale so I don’t think it is too much of an issue (+ in Tournayre et al. 2020 they showed that the percentage of bat reads had no effect on the number of occurrences detected). Would be interesting that you test it with your data to see if you get the same signal or not (although probably less stat power as it would be just galan vs zeale…?).

L651: here you could emphasize that it’s very common (if not always the case?) in metabarcoding diet study; it’s not that you did something wrong, it’s just incredibly hard to reach the plateau.

L662: how would you assess alterations in insect populations owing to anthropogenic pressures based solely on metabarcoding studies?

L668: add refs for the metabarcoding studies (e.g. Vallejo et al 2019, Vallejo et al 2023)

L686: presumably* prey

L716: It would be good to mention secondary predation issue somewhere (metabarcoding cannot distinguish the prey from the prey of the prey): it surely inflates diversity.

L735: I think it would be very valuable to make an analysis based on the landscape composition...

L743: I don’t know if I just misunderstood the sentence, but if not please be careful in your claims because you don’t have prey availability data. It sounds like you’re circling back your data here.

L747: suggests instead of indicates (again you don’t have prey availability to support your claim)

L751: Any idea why it is the opposite pattern in the Basque country (Vallejo et al 2023)?

L755: I know my comment is redundant but I’m not comfortable with the “two third of bat individuals”, you don’t know if your pellet belonged to one or several individuals - especially as adult individuals and mother/offspring can stay very close to each other. The latter would of course have an impact late summer when the young starts to hunt.

L817: would be good to see on the map :)

L881: And does ME also preys on endangered insect species? I so agree that bats are invaluable allies, but they probably also prey on endangered species. It would be nice if you could look into your data, especially as you are talking about how the decline of insect might affect ME in the Intro.

Table S1 and S2: Please complete the caption: “ME” = Myotis emarginatus, what are “CD” and “ED” column (I assume their the codified date for analyses)? What’s the difference between “NumberMEAdults” and colony size? Is colony size ME+RhiFer?

Again Table S3: Please complete the caption: what do the Column names mean, same for color code (why some species are highlighted in blue, red, orange)

Table S4: Please complete the caption: color code?

Reviewer #2: Major comments: Vescera et al. presents a study that describes the spatio-temporal variation in the diet of M. emarginatus, and then makes inferences about foraging behavior from bat diet. The laboratory techniques used for this study seem to be very thorough and I think using life history data from prey species to infer predator behaviors is fascinating! However, I do have some serious concerns about the authors’ overextension of their results and, by association, some of the major claims of their paper.

First, I think the emphasis on spatial variation is a bit overplayed. According to IUCN, this species ranges across most of Europe and parts of Africa and Asia. Your field sites seem to be an average of 50-75 km apart and only exist in their northern range. This to me is not spatially varied, especially as no information is provided in the Methods that discusses the variations in the landscapes at each of these data points (this is not seen until the Discussion). Focusing on the novelty of your work as the northern-most extreme for this species would be more accurate – which you somewhat allude to in Lines 712-714.

Second, temporal variation typically represents several time points throughout the year or life stages of an animal. With hibernating bats, the yearly active period is truncated in parts of the world due to hibernation, which can limit the number of time points to look at diet. However, bats are active on the landscape outside of the breeding season of June-August that has been identified here in this study. Therefore, to accurately assess temporal variation, bat diet should also be sampled outside of the breeding season if a study aims to look at actual temporal variation in bat diet. This study only focused on three time points during the breeding season (although even the temporal “sessions” are not explicitly defined, only in the Discussion does it say that each are explicitly a single month and align with specific periods of the breeding season). Additionally, this study only collected one season of data, which is not an accurate metric for temporal variation. I think it is an overstatement that this study looks at temporal variation and instead should narrow their inference to breeding season variation. Still incredibly important and is also more accurate given the presented data.

Third, the authors use previously published literature on prey species behaviors to try and characterize the foraging and prey hunting patterns of M. emarginatus. I think the idea behind this is really impressive and innovative, and I applaud the authors’ efforts. However, I think the authors need to measure their claims a bit about where bats are foraging. Explicitly stating that bats traveled and foraged at cowsheds, when your only evidence for this is that they consumed flies that contained DNA of cows, is extreme as the authors do not provide any additional evidence from this study to support their claim. How do the authors know that bats are foraging on these insects from cowsheds? From scanning the literature, flies can disperse several kilometers from their origin, meaning that bats may not be necessarily taking these pests from cowsheds (Nazni et al. 2005, Kjærsgaard et al. 2015). Therefore, some serious alterations in the language used to describe this potential relationship are needed. I recommend removing the focus on cowsheds from the title as well unless you introduce evidence that definitely shows bats are in cowsheds (like telemetry data, acoustic data, or capture data).

Lastly, throughout the manuscript, the authors refer to their samples as collected from individual bats. While the methods state that efforts were made to collect feces some distance apart so that it was likely from another bat, bats alter position in the roost all the time so there is no way that this method confirms data collection from individuals. If there was genetic analysis done to show that fecal samples were confirmed to be from individuals, that information needs to be included in the main manuscript. As such, language and inference needs to be adjusted to reflect that samples are from colonies and not individuals, so behavioral inferences can only be made on a colony level, not the individual.

Given these concerns, I suggest that the authors narrow the narrative for this manuscript to looking at breeding season variation in the northernmost point of the range of M. emarginatus. I think that is a more accurate representation of the results and is also still compelling. In the discussion, the authors could then provide inference about the foraging ecology of this species based on the diet, without stating that their study shows that bats are foraging in specific areas and using particular hunting strategies (because your study does not show these things). A very similar study recently published by Vallejo et al. in Mammalian Biology did a great job of not over-extending their results but still making inferences about foraging; Vescera et al. could do something similar that still shows how cool this information is without overdoing it.

As a general note, the grammar and language in this manuscript needs to be reviewed for conciseness, correctness, and for confusing sentence structure. In the line comments, I have highlighted areas I noticed that need assistance. However, I recommend the authors closely proofread their revision or ask for assistance with proofreading the English used in this manuscript in future versions.

Cited literature:

Kjærsgaard, A., Blanckenhorn, W. U., Pertoldi, C., Loeschcke, V., Kaufmann, C., Hald, B., ... & Bahrndorff, S. (2015). Plasticity in behavioural responses and resistance to temperature stress in Musca domestica. Animal Behaviour, 99, 123-130.

Nazni, W. A., Luke, H., Rozita, W. W., Abdullah, A. G., Sa'diyah, I., Azahari, A. H., ... & Sofian-Azirun, M. (2005). Determination of the flight range and dispersal of the house fly, Musca domestica (L.) using mark release recapture technique. Tropical biomedicine, 22(1), 53-61.

Vallejo, N., Aihartza, J., Olasagasti, L., Aldasoro, M., Goiti, U., & Garin, I. (2023). Seasonal shift in the diet of the notched-eared bat (Myotis emarginatus) in the Basque Country: from flies to spiders. Mammalian Biology, 1-13.

Line comments:

Line 27: reword to “…conservation, as it will aid our understanding of foraging behavior plasticity in response to reduced insect populations.” The way it is currently worded is cumbersome and could be refined to make your point clearer.

Line 28: reword to “Despite the global decline in insects,…” so it is more of standalone thought

Line 29: The English in the first part of this sentence needs a little work. Try something like “Past research has bypassed the potential of European bats to regulate such pest populations, despite their economic relevance.” This will also be important to adjust as there is a lot of work looking at pests in bat diet, so you will need to fix this sentence to make it more specific to your system.

Line 30: Remove “thus”.

Line 31: add “the” before “diet composition”.

Line 31: See my major comments above - you didn’t actually look at foraging behavior or pest hunting – you looked at diet and spatio-temporal variation. Adjust this sentence to show what your study actually focused on.

Line 32: Would you say M. emarginatus is a habitat specialist? Try to streamline this part of the sentence: “of a bat species originally favoring closed or semi-open habitats”, as it’s quite chunky and starts to seem like a run-on sentence.

Line 38: There is no data presented in this manuscript placing bats at cowsheds. Rephrase to say that they may be using cowsheds

Line 41: rephrase “~50% of taxa occurrences concerned these flies.” Do you mean “these flies accounted for almost 50% of Dipteran occurrences”? Rephrase to make it more interpretable.

Line 42: Are you inferring that bats are switching their diet in the middle of a single year due to decreases in insect populations? You do not provide any evidence to support this claim and the data you do provide are not comprehensive enough to support this statement. Revise to something more relevant to the data you present, maybe along the lines of breeding season needs this far north or something.

Line 45: This sentence seems like a random addition and comes after what seems like your concluding statement. Can you rephrase this part of the abstract to talk about pest suppression as a whole, then maybe a sentence that talks about livestock pests and plant pests? The end of the abstract is disorganized in its current form.

Line 53: This is getting slightly outdated – use Simmons and Cirranello 2023 (batnames.org). Its up to almost 1500 species now

Lines 53-54: Poor sentence structure. Removing the part about European bats and placing it in its own sentence would be better.

Line 55: remove “hence”

Lines 54-59: It seems that this information here is needed to demonstrate that we need bat diet data to help with bat conservation. I think you should trim this down to one sentence or two at most, because your paragraph oscillates between talking about bats, talking about insects, and then talking about bats again when it seems that the goal of this paragraph is to talk about bats, their diet, and how diet can inform conservation. Therefore, trim some of this information about insects down so that the paragraph mainly focuses on bats.

Line 66-78: A lot of your sentences start with connectors. These are really useful in moderation but when used too frequently they seem unnecessary. Try reading through and see where you can remove some of these connectors to help with sentence flow.

Line 75: “would be”? Would be consumed if what happened?

Line 77: remove “as such”

Line 77-78: I like this line of thinking!

Line 83: reword “It was already shown…” to “Previous work shows..”

Lines 83-85: These two sentences can be combined to add some varying sentence structure and remove extra words.

Line 86: “In regard to” instead of “As regards…”

Line 88: reword to “Most recently, a field experimental study…”

Line 90: “are threatened by climate change”

Line 91: “menace” is fairly emotive. You could try something like “Nearly 60% of European forests are threatened by climate change, with insect outbreaks accounting for 26% of potential damage”.

Line 94: “Albeit” is used incorrectly here. References 34 and 35 are not studies on bat regulatory action

Line 97-98: Is “livestock actual welfare regulations” a specific term? If not, remove “actual”.

Line 102: both what? You’ve provided three potential options here; both refers to just two.

Line 108: reword to “…arises from hunting flexibility and its association with trophic niche breadth.”. “the latter” doesn’t quite work in this sentence currently and comes off as unnecessary words.

Line 110: “thus” is unnecessary here.

Line 111: “hence” is unnecessary here. Rewording to “Bat species that display various hunting strategies can feed on more diverse prey and use a wide range of foraging grounds, ultimately decreasing their vulnerability to environmental heterogeneity.” Would help reduce excessive wordage and with directing the message of your sentence. Also, by heterogeneity, do you mean environmental changes? Many wildlife species require habitat heterogeneity, so this may not be the best word choice for this sentence.

Line 112: Shouldn’t you introduce the different types of hunting styles before stating that bats will use various hunting strategies which reduces their vulnerability to environmental heterogeneity?

Line 116: favored by who? Bats? Or do you mean conserved?

Line 117: “also” is not necessary in this sentence.

Line 118-119: What does the first part of this sentence mean? A close match in space and time? What does this have to do with whether their pests are considered prey?

Line 127: “increased invasion risk of pests”… This part of the sentence does not make sense with the beginning of the sentence… what are you trying to say exactly?

Line 143: This study does not employ a wide spatio-temporal scale. The spatial replication is restricted to part of Belgium and the temporal scale is across three time points in the breeding season. The language about this scale needs to be adjusted throughout the manuscript.

Line 155: I think you can say that you will discuss the potential hunting strategies and foraging habitats used by M. emarginatus. You don’t explicitly determine these behaviors given the current data, but you can have a section that discusses the potential strategies used by bats.

Line 165-169: This sounds like content for the Discussion.

Line 189: Why are you abbreviating to ME now instead of earlier in the manuscript?

Line 192: Need to explicitly define the temporal limits of each session. Also need to provide biological reasoning for why you delineated the breeding season into these sessions. You state in the discussion that these loosely align with breeding, lactation, and parturition periods but need to state here.

Line 196: why does it matter that R. ferrumequinum co-roosts with this species?

Line 197: Reformat references.

Line 204: Remove “being” – not needed in the sentence.

Line 205: Product information for the DNA decontaminant.

Line 207: So because this is under-roost sampling that cannot be traced back to individuals, you can’t really make claims like Line 38 about 55% of females foraging in cowsheds. You have no way of knowing that these fecal samples didn’t all come from a few bats as you do not discuss testing the bat DNA of the feces.

Line 239: What do you mean “to trace back samples’ origin” – like to see what bat it came from? What species? Further explanation is needed.

Line 258: What are ASVs?

Line 282: You need to state here or somewhere in the manuscript the number of samples that were analyzed from each site and from each session.

Line 290: You can’t say this based on your proposed methods, which state that you did not do any DNA testing confirming that fecal samples came from individual bats.

Line 313: re-arrange to “We used the number of taxa identified per order in each sample as the response variable.”

Line 334: Should try to stay consistent with terminology – spiders vs. Aranae. Also, what is meant by closed-landscapes? Like closed-canopy?

Line 356: What are the criteria for denoting a species as a minor or major pest? None listed.

Line 366: This sentence needs to be incorporated somewhere as it is a single sentence that can’t stand alone as a paragraph.

Line 402: Which level of taxonomy is this sentence referring to? You’ve talked about several different taxonomic levels so far, so it is unclear which is being discussed here. If stated previously, I suggest refreshing the reader’s mind and adding a statement here about which taxonomic levels were analysed in this section.

Line 410: I feel like this makes sense as the variation in temporal data collection points is very limited. Could be a good discussion point.

Line 415: reword to “In regard to sample coverage curves,”

Line 417: “overall good capturing” is not research-grade English and I suggest rewording.

Line 420: These sentiments provide evidence that your data does not provide much in the way of spatial variation.

Line 499: “In regard to variation in diet composition”. “As regards” only works in certain settings, and I don’t think it fits here. Better yet – remove “In regard” and try “Our NMDS plot showed that locality was not a major driver of variation in diet composition (Fig 6).” There are a lot of areas throughout the document that can remove unnecessary words to help with the directness of phrasing and increase reader comprehension.

Lines 516-518: What does this statistic tell the reader? Just that these two things are correlated, which does not provide sufficient evidence to show that bats are foraging for these spiders in forests.

Line 535: You cannot say 163 individual bats because from the methods, it does not sound like samples were collected from individuals – they were collected from under roosts. If you provide any genetic analyses showing that fecal samples are from individual bats, then that needs to be presented. But in its current form, you cannot state anything about samples from individuals.

Line 537: Same thing here about discussing individuals.

Line 558: Same thing here about discussing individuals.

Line 582: 55.38%

Line 584-585: What is this statistic supposed to demonstrate exactly? Similar to my comment above, it is just looking at correlation without any additional data to make it biologically meaningful in this situation. Would cow presence be more useful in this situation? As you could show that fly presence is correlated with cow presence at least? Regardless, this is still not evidence for the foraging in cowsheds claim.

Line 607: The temporal and spatial coverage of your samples are not extensive, I suggest rephrasing.

Line 614: No supporting evidence for this claim.

Line 616: No supporting evidence for this claim.

Line 617: I suggest a more appropriate term for research instead of “great”. Maybe “thorough”? Or even “excellent”? “High-quality”?

Line 649: I think this is a great way to frame your findings! You looked at the regional diet of ME, which is important because this is the northernmost point of their range. That claim is actually supported by the data you present here and could make for a compelling story.

Line 651: Each sample, not each bat because there is no statement or data showing that each sample came from an individual bat – only that you tried to get them from individuals based on where they were roosting. That is not the same as definitively collecting samples from individuals.

Line 701: individual samples, not bats.

Line 712-714: This is an important point as it shows that your sites do not really represent spatial variation relative to the range of this bat species.

Line 721-723: Why is this single sentence a separate paragraph? There are several cases of this throughout the manuscript that need to be addressed as a single sentence typically cannot be a paragraph on its own.

Line 755: Provide a percentage for how often cow DNA was detected in samples with flies.

Line 756: Cannot say use of cowsheds for foraging as you have no evidence. That may be where the cows the flies prey upon are, but flies can disperse over several kilometers, meaning they are not limited to cowsheds.

Lines 818-821: Why is this a separate paragraph? It’s not developed enough to stand on its own in its current form.

Lines 822-823: There is no evidence confirming foraging in these areas. You can say that they may forage in these areas based on captured prey, but as I’ve mentioned previously these fly species are not restricted to only occurring in cowsheds and there is huge variety in where spiders and lepidopterans may live. Further, you do not provide much context for the surrounding landscapes, so it is hard to infer if these bats are selecting these specific foraging regions you mention or if they are just foraging based on what’s available.

Line 823: As you state here, you inferred these claims. That is fine, but you need to be up front that these are inference-based claims, not data-driven claims. I do think there is room with your data to provide such suggestions that bats may forage in cowsheds and these other areas, but you currently do not have the data to be so specific with your claims.

Line 838: What do these numbers next to minor and major mean? Are those the number of taxa in each category?

Reviewer #3: Review of the manuscript PONE-D-23-33429 “All-you-can-eat buffet: A spider-specialized bat species (Myotis emarginatus) turns into a pest flies’ eater in cowsheds. In this investigation, the authors investigated diet composition, its spatio-temporal variations, foraging behaviour, and pest hunting of Myotis emarginatus from six location in Wallonia,Belgium. Overall, I found the manuscript too lengthy with very confused objectives and methods. I think the most interesting result found by the authors is the fact that bats complement or shift their diet with diptera in the most energetic demanding phase of reproduction. I recommend the authors to reduce the length of the manuscript by focusing in this main result. I have made some recommendations below.

Introduction

I find the introduction quite extensive and jumpy. The authors use a broad number of terms that are not properly defined, which makes the reading confusing. In the text, global warming is largely mentioned. Nevertheless, the reduction in entomofauna due global warming goes out of the scope of the investigation, as the authors did not take any long-term measurement to establish a metric of this phenomena. I suggest reduce the length of the introduction by eliminating all the information regarding global warming and entomofauna reduction.

Specific observations

L 43-44. What do you mean the cowsheds act as ecological traps? I think this does not make much sense as is presented in this part of the manuscript.

L 59. “and food provisioning”

L 76-78. I do like this idea showing bats as samplers of biodiversity. Nevertheless, I would discard this idea because in such case the sampling would be biased by the preferences of prey by the different species composing the order.

L 84. Is this species a pest in the pine forest?

L 96. By disturbing feeding of “livestock” and their ability…

L 97. I can not understand how S. calcitrans could reduce meat production?

L 103 What topic?

L 105-106. If bats do not restrict their foraging activity to a single habitat, how cattle-related “habitats” can be so hazardous for them (L 97-100)

L108 What “hunting flexibility” mean?

L How are “hunting strategies” and “hunting stiles” (L112) alike or not?

L 111-112 “hence decreasing their vulnerability to environmental heterogeneity”. What do you mean with this? Please clarify

L 111, 114. What do you mean by foraging grounds?

L 127-129. Yet, this was not the main goal to study in your investigation.

L 165-169. This information should be put in the discussion section of the manuscript, not here.

Methods

The methods section is difficult to follow as it has many undefined variables that the authors used for the different analyses. The methods are also jumpy and poorly understandable. Most of the analysis do not have any support to fulfill the objectives pursued by the investigation. I suggest to present the main objectives and the analysis related to these objectives in one paragraph and then assess them in detail in the order that were presented in the following paragraphs.

Some information about the different breeding colonies is missing. Are those sites surrounded by vegetation or rather immerse in cities? If that’s so, which kind of vegetation was? How far are they separated each from the other? I think this information is relevant because is related to prey availability which is related to the main objectives of the investigation.

I would reduce the number of acronyms used because is difficult to follow them along the manuscript.

Please add package citations among the methods.

Specific observations

L 188-189. Myotis emarginatus must be abbreviated as M. emarginatus not ME.

L 192-193. It would be easer naming the sessions as the months (June, July, August) instead of S1, S2 and S3

L 285-295. Were all those metrics calculated for the whole monitoring? Specify

L 299-300. What was the purpose of doing these calculations for each order?

L 307. What was the temporality you are talking about? wasn’t the study done only during the breeding season (L188)

L 309. “localities/sessions” is the session each one of the visits you performed? the diagonal between those variables indicates that they were nested in the analyses?

L 310. What was the sample coverage?

L 313. When you refer to taxa you mean species fully identified?

L 314-315. “When locality/session was introduced as a fixed factor, we considered session/locality as a random factor” to what end? Explain

L 319. “Then, we tested the impact of spatial and temporal variation” Which quantifiable variables were those?

L 339-342. “To get insights into spiders’ ecology” what this does mean? Also, what was the end of characterizing functional groups of spiders related to the main objective of the investigation?

L 343. Does this mean you performed a spiders monitoring? How was this conducted? This is not presented in any part of the manuscript.

L 348-351. I do not understand this at all. What was the end of doing so? Which were the response and explanatory variables? What was the nature (i.e., numerical or categorical) of the data? If some variable was categorical, how many levels does it had?

L 353. Eliminate repeated word.

L 366. Statistical differences are stablished when alpha and not p value are < 0.05.

Discussion

L 612-613. It should be parturition and lactation period.

L 614-616. I wonder why the authors are sure to assert that the house flies are hyper abundant in the cattle sheds since they did not take any measurement during the performance of the study.

L 617-662. I would reduce this section of the manuscript.

L 618. While I agree that metabarcoding is a good technique to assess diet in bats, I am not sure that insectivorous bats would be excellent natural samplers of biodiversity. As I stated above, prey consumed by bats may depend of different preferences of the species (many of them linked to the animals’ physiology). Also, because bats travel a variety of distances and habitats in a single night to forage, we could not be able to associate concrete areas holding such diversity. This may difficult the interpretation of the diversity patterns related to the ecosystems.

L 617. I think that the discussion of the method should be moved to the last

L 620-621. Were those studies performed in the same area and temporality? Differences in prey consumed could be resulted just because differences on these characteristics.

L 660-662. See my previous comment on this regard.

L 724-737. This is the reason because I asked before for the characteristics of the shelters to be described in the methods section. Differences in the areas surrounding the shelters may explain differences in the food items presents in the pellets of individuals.

L 885-881. I would reduce this section of the manuscript.

L 882-901. I would reduce this section of the manuscript.

6. PLOS authors have the option to publish the peer review history of their article (what does this mean?). If published, this will include your full peer review and any attached files.

Reviewer #1: No

Reviewer #2: No

Reviewer #3: Yes

---

## [Author Response · Author response to Decision Letter 0]

16 Feb 2024

All-you-can-eat buffet: A spider-specialized bat species (Myotis emarginatus) turns into a pest fly eater around cattle

Editor answer:

Dear Dr. Vescera,

Thank you for submitting your manuscript to PLOS ONE. After careful consideration, we feel that it has merit but does not fully meet PLOS ONE’s publication criteria as it currently stands. Therefore, we invite you to submit a revised version of the manuscript that addresses the points raised during the review process.

We look forward to receiving your revised manuscript.

Kind regards,

Camille Lebarbenchon

Academic Editor

PLOS ONE

Dear Dr. Camille Lebarbenchon, 

Thank you very much for having considered this manuscript for review and for the selection of skillful reviewers. They all made relevant comments which hopefully helped me bring this work to the next level. I took into account most of their wise suggestions to improve this manuscript in terms of language, scientific accurateness and conciseness. Thank you also for the advice you made to deposit my protocols on a designed platform. I already thoroughly described my methods in the manuscript and using supporting information. Any reader should therefore find all necessary experimental details in the present submission. However, I will consider the opportunity provided by PLOS ONE for publishing peer-reviewed Lab Protocol articles. 

After having addressed all points raised during the review process, I now feel optimistic about the quality of this improved manuscript and about its compliance with the PLOS ONE’s publication criteria.

Thank you again for the time devoted to the submission of this research. 

Best regards, 

Chloé Vescera

Journal Requirements:

I double-checked all file names so that they meet PLOS ONE's style requirements.

I modified it accordingly. 

3. We note that you have referenced (Brinkmann, unpublished) on page 29, which has currently not yet been accepted for publication. Please remove this from your References and amend this to state in the body of your manuscript: (ie “Bewick et al. [Unpublished]”) as detailed online in our guide for authors

I formatted this reference such as detailed in the guide for authors. 

4. We note that [Figure 1] in your submission contain [map/satellite] images which may be copyrighted. All PLOS content is published under the Creative Commons Attribution License (CC BY 4.0), which means that the manuscript, images, and Supporting Information files will be freely available online, and any third party is permitted to access, download, copy, distribute, and use these materials in any way, even commercially, with proper attribution. For these reasons, we cannot publish previously copyrighted maps or satellite images created using proprietary data, such as Google software (Google Maps, Street View, and Earth). For more information, see our copyright guidelines: http://journals.plos.org/plosone/s/licenses-and-copyright.

All maps displayed in this manuscript have been constructed using QGIS tools by the authors based on shapefiles freely accessible on the LifeWatch database (https://maps.elie.ucl.ac.be/lifewatch/ecotopes.html#). This platform offers quality maps and supports “Open e-Data for Biodiversity”. This data complies with the CC BY 4.0 license which enables to:

- Share — copy and redistribute the material in any medium or format for any purpose, even commercially.

- Adapt — remix, transform, and build upon the material for any purpose, even commercially.

I therefore included the sentence mentioned above (“Reprinted from [ref] under a CC BY license, with permission from [name of publisher], original copyright [original copyright year].”) in the figure caption but I did not complete the Content Permission Form as the original copyright holder explicitly states that the content is free to use.

Reviewers' comments:

Reviewer's Responses to Questions

Comments to the Author

1. Is the manuscript technically sound, and do the data support the conclusions?

Reviewer #1: Partly

Reviewer #2: No

Reviewer #3: Partly

2. Has the statistical analysis been performed appropriately and rigorously?

Reviewer #1: N/A

Reviewer #2: No

Reviewer #3: Yes

3. Have the authors made all data underlying the findings in their manuscript fully available?

Reviewer #1: No

Reviewer #2: Yes

Reviewer #3: Yes

4. Is the manuscript presented in an intelligible fashion and written in standard English?

Reviewer #1: Yes

Reviewer #2: No

Reviewer #3: Yes

5. Review Comments to the Author

Reviewer #1: 

Summary:

This study investigates the diet of the Geoffroy bat (Myotis emarginatus, ME) in several sites at different times during the breeding season. Fecal pellets were collected beneath roosting sites and analyzed with two COI metabarcoding that were further compared based on their respective detection success. The authors had several objectives: exploring the spatial and temporal variation of the diet and assessing the hunting strategy and foraging habitat used by ME by looking at prey ecological trait and habitat. The obtained results were also used to determine if the processed samples were a sufficient representation of the total Geoffroy bat diet.

General Comments:

I enjoyed this interesting work. The necessity to study the diet of Geoffroy bats in combination with the landscape is well explained. The methods are overall well described (there is some missing information, and some stats should be clarified), and most interpretations are supported by the results (in my opinion a few claims should be toned down in the Discussion). I also suggest including a section on secondary predation (e.g. it preys on spiders that prey on insects and/or other arthropods) in the discussion as it could have a big impact on diet richness estimation. Generally, I found the manuscript lengthy and sometimes hard to follow, so I believe the manuscript would benefit from being condensed a little bit (filler and unnecessary words could also be removed, e.g. of unnecessary words: “as a matter of fact”).

Note: I haven’t been able to access PRJNA1019357 (it was not found in SRA).

Thank you very much for this insightful and thorough review on my paper. These comments proved helpful to improve this work. As suggested, I added a section on secondary predation in the first part of the Discussion. I also removed some unnecessary sections, i.e. the ones that did not take part in the main story of this manuscript. I further refined this work by clarifying some stats and making explicit links between the goals, what has been done to achieve them and what were the final outputs and their interpretations. As such, I hope that the flow and clarity of the revised version have been improved. 

Regarding the access to the sequence data related to this work, it will be allowed as soon as the paper will be released. This is why the given code is currently not working. However, I ensure you that raw sequencing data has well been deposited on this database. Here is a link created for reviewers, which provides read-only access to metadata: https://dataview.ncbi.nlm.nih.gov/object/PRJNA1019357?reviewer=lhpdp8f56thb7hjcnhr78iacs1. Note that this URL will expire when this BioProject is publicly-released.

Abstract

L46: You highlighted that ME prey on some pest species and could potentially be considered a pest regulator. We can’t say from your data whether they have an impact on pest population as we just know that the species were detected in the feces.

I fully agree and modified it accordingly by replacing the concept of “regulator” by “consumer”, the latter being only associated to the action of eating. 

Introduction

L165-169: This section should not be in Introduction.

I removed this section and incorporated it in the Discussion.

Note: There is an arbitrary writing rule on using numerals for numbers 10 and greater and spelling out numbers one through nine.

I modified all number—except for numerals used for measurements—accordingly, thank you for the tip. 

Material & Methods

L193: Why did you choose to refer to S1, S2, S3? I would recommend keeping the month instead of sessions (June/July/August or gestation/lactation/post-lactation). That would be so helpful to understand your data (linking your observations to the bat life cycle, energetic needs etc).

This indeed would be so much clearer. As such, I replaced each session by the corresponding month.

Figure 1: It is a pretty map, but as your paper focuses on landscape (forest, cowsheds) it would be more useful for us to know the landscape surrounding the sampling sites (forest, farm land etc.: CORINE land cover map?) instead of Wallonia regions.

Thank you for the advice, this would actually make much more sense in the context of this paper. I still kept this initial map to get a quick view of the location of the breeding sites in Wallonia but I placed it as supplementary material (S1 Fig). I then created two new representations: (1) a map depicting nine landcover classes within a 10 km radius around each bat maternity roost (Fig 1) and (2) a graph displaying the relative proportion of these landcover classes around each maternity roost (Fig 2). At the Walloon scale, we have a more precise land cover mapping than CORINE land cover map, i.e. the LifeWatch database (available here: h

---

## [Editor Report · Decision Letter 1]

27 Mar 2024

All-you-can-eat buffet: A spider-specialized bat species (Myotis emarginatus) turns into a pest fly eater around cattle

PONE-D-23-33429R1

Dear Dr. Vescera,

We’re pleased to inform you that your manuscript has been judged scientifically suitable for publication and will be formally accepted for publication once it meets all outstanding technical requirements.

Kind regards,

Camille Lebarbenchon

Academic Editor

PLOS ONE
---

## [Editor Report · Acceptance letter]

26 Apr 2024

PONE-D-23-33429R1 

PLOS ONE

Dear Dr. Vescera, 

I'm pleased to inform you that your manuscript has been deemed suitable for publication in PLOS ONE. Congratulations! Your manuscript is now being handed over to our production team.

Kind regards, 

on behalf of

Dr. Camille Lebarbenchon 

Academic Editor

PLOS ONE